# VARIATIONAL DOMAIN ADAPTATION

## ABSTRACT

This paper proposes *variational domain adaptation*, a unified, scalable, simple framework for learning multiple distributions through variational inference. Unlike the existing methods on domain transfer through deep generative models, such as StarGAN (Choi et al., 2017) and UFDN (Liu et al., 2018), the variational domain adaptation has three advantages. Firstly, the samples from the target are not required. Instead, the framework requires one known source as a *prior* $p(x)$ and binary *discriminators*, $p(\mathcal{D}_i|x)$, discriminating the target domain $\mathcal{D}_i$ from others. Consequently, the framework regards a target as a *posterior* that can be explicitly formulated through the Bayesian inference, $p(x|\mathcal{D}_i) \propto p(\mathcal{D}_i|x)p(x)$, as exhibited by a further proposed model of *dual variational autoencoder* (DualVAE). Secondly, the framework is scablable to large-scale domains. As well as VAE encodes a sample $x$ as a mode on a latent space: $\mu(x) \in \mathcal{Z}$, DualVAE encodes a domain $\mathcal{D}_i$ as a mode on the dual latent space $\mu^*(\mathcal{D}_i) \in \mathcal{Z}^*$, named *domain embedding*. It reformulates the posterior with a natural paring $\langle, \rangle : \mathcal{Z} \times \mathcal{Z}^* \to \mathbb{R}$, which can be expanded to uncountable infinite domains such as continuous domains as well as interpolation. Thirdly, DualVAE fastly converges without sophisticated automatic/manual hyperparameter search in comparison to GANs as it requires only *one* additional parameter to VAE. Through the numerical experiment, we demonstrate the three benefits with multi-domain image generation task on CelebA with up to 60 domains, and exhibits that DualVAE records the state-of-the-art performance outperforming StarGAN and UFDN.

## 1 INTRODUCTION

*"...we hold that all the loveliness of this world comes by communion in* Ideal-Form. *All shapelessness whose kind admits of pattern and form, as long as it remains outside of Reason and Idea, is ugly from that very isolation from the Divine-Thought."* — Plato (427 – 347 bc)

Agents that interact in various environments have to handle multiple observation distributions . Domain adaptation (Bengio, 2012) is a methodology employed to exploit deep generative models, such as adversarial learning (Goodfellow et al., 2014) and variational inference (Kingma & Welling, 2013), that can handle distributions that vary with environments and other agents. Further, multi-task learning and domain transfer are examples of how domain adaptation methodology is used. We focus on domain transfer involving transfers across a distribution between domains. For instance, pix2pix (Isola et al., 2017) outputs a sample from the target domain that corresponds to the input sample from the source domain. This can be achieved by learning the pair relation of samples from the source and target domains. CycleGAN (Zhu et al., 2017a) transfers the samples between two domains using samples obtained from both domains. Similarly, UNIT (Liu et al., 2017), DiscoGAN(Kim et al., 2017), and DTN(Taigman et al., 2016) have been proposed in previous studies.

However, the aforementioned method requires samples that are obtained from the target domains, and because of this requirement, it cannot be applied to domains for which direct sampling is expensive or often impossible. For example, the desired, continuous, high-dimensional action in the environment, intrinsic reward (e.g., preference and curiosity) and the policy of interacting agents other than itself cannot be sampled from inside, and they can only discriminate the proposed input. Even for ourselves, the concept of beauty or interest in our conscious is subjective, complex, and difficult to be sampled from the inside, although it is easy to discriminate on the outside.

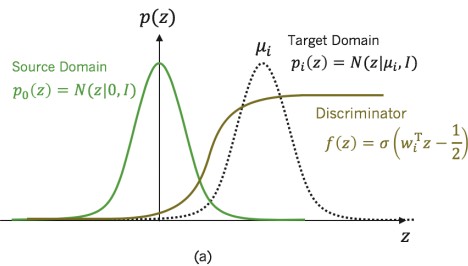 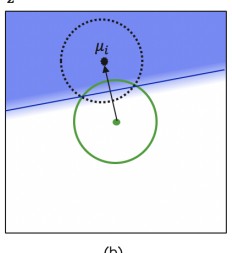 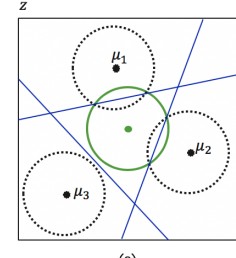

Figure 1: The key concept of variational domain adaptation. **a)** Given the proposal drawn from the prior, the discriminator discriminates the target domain from the others. Each domain is posterior for the prior $\mathcal{N}(z|0, 1)$; further, the distribution in the latent space is observed to be a normal distribution using the conjugate likelihood. **b)** Domain transfer is represented by the mean shift in the latent space. **c)** Domain embedding: After training, all the domains can only be represented by vectors $\mu_i$.

In this study, we propose *variational domain adaptation*, which is a framework for targets that pose challenges with respect to direct sampling. One solution is *multi-domain semi-supervision*, which converts the problem to semi-supervised learning, thereby making is possible to perform variational inference. In this supervision, a source domain is regarded as a *prior* $p(x)$ and a target domain is considered to be a *posterior* $p(x|\mathcal{D}_i)$ by referring to the label given by a supervised *discriminator* $p(\mathcal{D}_i|x)$ that distinguishes the target domain from others. Our model imitates the behavior of the discriminator and models the target domain using a simple conclusion of the Bayesian theorem, $p_\theta(x|\mathcal{D}_i) \propto p_\theta(\mathcal{D}_i|x)p_\theta(x)$. The end-to-end learning framework also makes it possible to learn good prior $p_\theta(x)$ with respect to all the domains. After the training was completed, the posterior $p_\theta(x|\mathcal{D}_i)$ succeeded in deceiving the discriminator $p(\mathcal{D}_i|x)$. This concept is similar to rejection sampling in the Monte Carlo methods. Further, variational domain adaptation is the first important contribution from this study.

The second contribution from this study is a model of *dual variational autoencoder* (DualVAE), which is a simple extension of the conditional VAE (Kingma et al., 2014), employed to demonstrate our concept of multi-domain semi-supervision. DualVAE learns multiple domains in one network by maximizing the variational lower bound of the total negative KL-divergence between the target domain and the model. DualVAE uses VAE to model the prior $p(x)$ and an abstract representation for the discriminator $p(\mathcal{D}_i|x)$. The major feature of DualVAE is *domain embedding* that states that all the posteriors are modeled as a normal distribution $\mathcal{N}(z|\mu_i, \sigma^2)$ in the same latent space $\mathcal{Z}$ using the conjecture distribution of the prior. Here, $\mu_i$ is the domain embedding that represents the domain $\mathcal{D}_i$. This enables us to sample from $p_\theta(x|\mathcal{D}_i)$. Our major finding was that the discriminator of DualVAE was a simple inner product between the two means of domain embedding and the VAE output:

$$\log \frac{p_\theta(\mathcal{D}_i|x)}{p_\theta(\mathcal{D}_i)} = \log \int \frac{\mathcal{N}(z|\mu_i, \sigma^2)\mathcal{N}(z|\mu_\phi(x), \sigma^2)}{\mathcal{N}(z|0, I)} dz = \frac{\mu_i^T \mu_\phi(x)}{\sigma^2},$$

that acts as a natural paring between the sample and the domain. The probabilistic end-to-end model learns multiple domains in a single network, making it possible to determine the effect of transfer learning and to learn data that multi-domains cannot observe from sparse feedback. Domain embedding is a powerful tool and allows us to use VAEs instead of GANs.

The third contribution of this study is that DualVAE was validated for use in a recommendation task using celebA (Liu et al., 2015). In the experiment, using celebA and face imaging data obtained based on evaluations by 60 users, an image was generated based on the prediction of user evaluation and an ideal image that was determined to be good by multiple users. We demonstrated that the image could be modified to improve the evaluation by interpolating the image, and the image was evaluated using the domain inception score (DIS), which is the score of the model that has learned the preference of each user. We present the beauty inside each evaluator by simply sampling $p_\theta(x|\mathcal{D}_i)$. The DIS of DualVAE is higher than that of a single domain, and the dataset and code are available online.

## 2 RELATED WORK

The existing literature related to the domain transfer is based on the assumption that the samples are obtained from the target domain. For example, pix2pix(Isola et al., 2017) can output the samples from the target domain that corresponds to the input samples from the source domain by learning the pair relation between the samples of the source and target domains. CycleGAN (Zhu et al., 2017a), which differs from pix2pix, does not require sample pairs from both domains. Similarly, UNIT (Liu et al., 2017), DiscoGAN(Kim et al., 2017), and DTN(Taigman et al., 2016) also do not require sample pairs. Furthermore, because there are few cases in which samples from the source and target domains form a one-to-one pair in real world research after being extended to the conversion of one-to-many relationships, including BicycleGAN(Zhu et al., 2017b) and MUNIT(Huang et al., 2018).

Several studies were conducted to model multiple distributions in a semi-supervised manner. Star-GAN(Choi et al., 2017), UFDN(Liu et al., 2018), and RegCGAN(Mao & Li, 2018) are extensions of the aforementioned models and are frameworks that can convert the source domain samples into samples for various target domains with a single-network structure. However, the problem with these methods is associated with hyperparameter tuning, which arises from the characteristics of adversarial learning. DualVAE is a simple extension of a conditional VAE in a multi-domain situation. Conditional VAEs utilizes VAE for semi-supervised learning. Although the model is quite simple, it is powerful and scalable making it possible to learn multiple distributions with *domain embedding*. In fact, we demonstrated that DualVAE quickly converged for more than 30 domains without sophisticated hyperparameter tuning. In the experiment conducted in this study, $\mathbb{E}_\omega \left[ J(\theta|\omega) \right]$ was evaluated instead of $J(\theta|\hat{\omega})$ to demonstrate that our method required less hyperparameter tuning.

## 3 METHOD

### 3.1 PROBLEM DEFINITION

With regard to $n$ domains $\mathcal{D}_1, \ldots, \mathcal{D}_n$, and a sample $x$ on an observation space $\mathcal{X}$, the objective of unsupervised domain adaptation is to minimize the KL-divergence between the target distribution and the model, $D_{\mathrm{KL}} \left( p^{(i)}(x) \| p^{(i)}(x, \theta) \right)$, over all the domains $\mathcal{D}_i$. From the perspective of optimizing $\theta$, minimizing the KL divergence is equivalent to maximizing the cross-entropy. As $p^{(i)}(x, \theta) = p^{(i)}(x|\theta)p(\theta)$, the unsupervised domain adaptation can be formulated as a maximizing problem for the weighted average of cross-entropy over the domains:

$$\text{Maximize}_\theta : \quad J(\theta) = \frac{1}{n} \sum_{i=1}^n \gamma_i \, \mathbb{E}_{x \sim p^{(i)}} \left[ \log p_\theta^{(i)}(x) \right] + \gamma \log p(\theta), \quad (1)$$

where $p_\theta^{(i)}(x) = p^{(i)}(x|\theta)$, $\gamma_i \in [0, 1]$ is the importance of each domain $\mathcal{D}_i$ and $\gamma = \sum_{i=1}^n \gamma_i / n$. If $\gamma_i = 1$ for all the $i$'s, the objective function is simply the mean, and if $\gamma_i = 0$ for certain $i$'s, the domain $\mathcal{D}_i$ is ignored.

The difficulty arises from the fact that it is not possible to directly sample $x$ from $p^{(i)}$ $x$ can be directly sampled from the likelihood $p(\mathcal{D}_i|x)$. This challenge was the motivation for considering multi-domain semi-supervision.

### 3.2 MULTI-DOMAIN SEMI-SUPERVISION

*Multi-domain semi-supervision* assumes a prior $p(x)$ and models each the domain as a posterior $p^{(i)} = p(x|\mathcal{D}_i)$. As the Bayesian inference, we reformulate the cross-entropy $\mathbb{E}_{x \sim p^{(i)}} \left[ \log p_\theta(x|\mathcal{D}_i) \right]$ in Eq. (1) as follows:

$$\mathbb{E}_{x \sim p^{(i)}} \left[ \log p_\theta(x|\mathcal{D}_i) \right] = \int p(x|\mathcal{D}_i) \log p_\theta(x|\mathcal{D}_i) dx = \int \frac{p(\mathcal{D}_i|x)p(x)}{p(\mathcal{D}_i)} \log \frac{p_\theta(\mathcal{D}_i|x)p_\theta(x)}{p_\theta(\mathcal{D}_i)} dx$$

$$= \mathbb{E}_{x \sim p} \left[ \frac{p(\mathcal{D}_i|x)}{p(\mathcal{D}_i)} \log \frac{p_\theta(\mathcal{D}_i|x)}{p_\theta(\mathcal{D}_i)} \right] + \mathbb{E}_{x \sim p} \left[ \frac{p(\mathcal{D}_i|x)}{p(\mathcal{D}_i)} \log p_\theta(x) \right]$$

$$= \mathbb{E}_{x \sim p} \left[ f(\mathcal{D}_i|x) \log f_\theta(\mathcal{D}_i|x) \right] + \mathbb{E}_{x \sim p} \left[ f(\mathcal{D}_i|x) \log p_\theta(x) \right], \quad (2)$$

where $f(\mathcal{D}_i|x) = p(\mathcal{D}_i|x)/p(\mathcal{D}_i)$ and $f_\theta(\mathcal{D}_i|x) = p_\theta(\mathcal{D}_i|x)/p_\theta(\mathcal{D}_i)$. By letting $\gamma_i = p(\mathcal{D}_i)$, the objective is identical to:

$$J(\theta) = \underbrace{\mathbb{E}_{x \sim p, i \sim [n]} \left[ p(\mathcal{D}_i|x) \log f_\theta(\mathcal{D}_i|x) \right]}_{\text{discriminator}} + \underbrace{\mathbb{E}_{x \sim p} \left[ \log p_\theta(x) \right]}_{\text{prior}} + \underbrace{\gamma \log p(\theta)}_{\text{regularizer}}, \qquad (3)$$

where $[n]$ is a uniform distribution over $\{1, \ldots, n\}$ and $f(\bar{\mathcal{D}}|x) = \mathbb{E}_{i \sim [n]} [f(\mathcal{D}_i|x)]$. The first term is the likelihood from the discriminator; the second term is the prior learned by a generative model, including VAE; and the last term is the regularizer.

Because the equation is intractable, we use Monte Carlo sampling to estimate the function. During the estimation, we initially sample $x_1, \ldots, x_m$ from the prior $p(x)$ and subsequently obtain the binary labels $y_{ij} \in \{0, 1\}$ from each discriminator $y_{ij} \sim p(\mathcal{D}_i|x_j)$. Since the number of labels from supervises is $nm$, the situation that the sparse labels: $k << nm$ is considered. Further, some discriminators only provide parts of the labels. In the situation, the missing values are 0-padded: $y_{ij} = 0$.

$$J(\theta) \approx \frac{1}{n} \sum_{i=1}^{n} \sum_{j=1}^{m} y_{ij} \log f_\theta(y_{ij}|x_j) + \frac{1}{m} \sum_{j=1}^{m} \log p_\theta(x_j) + \bar{y} \log p(\theta), \qquad (4)$$

where $\approx$ indicates Monte Carlo estimation and $\bar{y} = \sum_{i=1}^{n} \sum_{j=1}^{m} y_{ij}/k$. In the limit of $n \to \infty$, the right side of the equation is identical to the left side.

### 3.3 DUAL VARIATIONAL AUTOENCODER (DUALVAE)

We extended the VAE for multi-domain transfer to demonstrate our concept of multi-domain semi-supervision. Our proposed model, *dual variational autoencoder* (DualVAE), models each domain $p_i(x)$ as a posterior distribution $p(x|\mathcal{D}_i)$ that is similar to that observed in a conditional VAE. Fig. 2 depicts the VAE and DualVAE graphical models.

The major feature of DualVAE is *domain embedding*, where all the domains and the prior share the same latent space $\mathcal{Z}$. For the prior distribution, $p(z) = \mathcal{N}(z|0, I)$ and $p(z|\mathcal{D}_i) = \mathcal{N}(z|\mu_i, \sigma^2 I)$, where $\mu_i \in \mathcal{Z}$ is an embedding and $I$ is a unit matrix in $\mathcal{Z}$. In the following, we denote $\sigma^2 I = \sigma^2$ without loss of generality. The domain $\mathcal{D}_i$ is characterized only by its embedding $\mu_i$. Here, $\mu_0$ is the embedding of the prior that can be assumed to be $\mu_0 = 0$.

Training DualVAE is virtually equivalent to simultaneously training $(n + 1)$ VAEs which share a parameter, including the prior. Using conjecture distribution for the prior $p(z)$, the posterior distribution is observed to be a normal distribution. Therefore, *all the posteriors are VAEs*. The joint distribution can be given as follows:

#### 3.3.1 VAE: THE PRIOR $p_\theta(x)$

A VAE (Kingma & Welling, 2013) is used to model the prior $p(x)$, a deep generative model that employs an autoencoder to model the hidden variable as random variable. The benefit of a VAE is that it can be used to model each distribution as a normal distribution in $\mathcal{Z}$, achieved by maximizing the variational lower bound of $\log p(x)$ as follows:

$$\log p_\theta(x) \geq \mathcal{L}_\theta(x) = \mathbb{E}_{z \sim q_\phi(\cdot|x)} \left[ \log p_w(x|z) \right] - D_{\text{KL}} \left( q_\phi(z|x) \| p(z) \right), \qquad (5)$$

where $\phi, w \in \theta$ is a parameter of the encoder and the decoder, respectively. The objective is to learn a pair of the encoder $p_w(x|z)$ and the decoder $q_\phi(z|x)$ to maximize $\mathcal{L}(x)$. $z$ acts as a prior $p(z) = \mathcal{N}(z|0, I)$.

The lower bound $\mathcal{L}_\theta(x)$ is derived using the reconstruction error and penalty term as the KL divergence between the model and the prior $p(z)$. Further, the gradient of the reconstruction term can be calculated using the Monte Carlo method, and because the construction term is the KL divergence between two normal distributions, it can be analytically calculated.

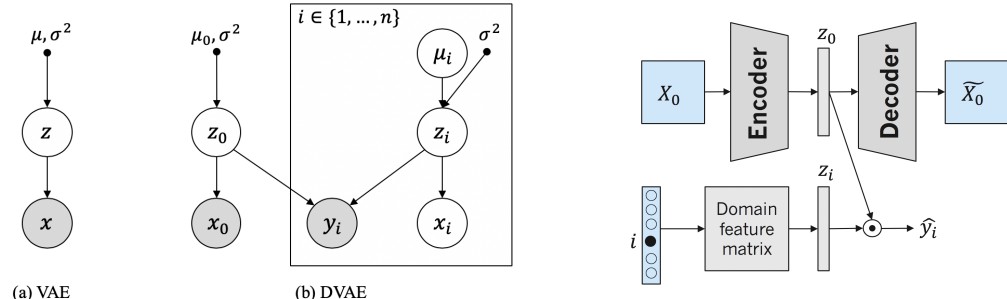

(a) VAE    (b) DVAE

Figure 2: **Left**: Graphical models of the probabilistic models of VAE and DualVAE. The gray and white circles indicate the observed variables and latent variables, respectively. Symbols without circles indicate the constants. Arrows between the symbols indicate probabilistic dependency (e.g., X generates Y). A rectangle with suffixes indicates a block, which comprises multiple elements. **Right:** The network structure of DualVAE. The label is structured as the inner product of latent $z_\theta$ and domain embedding $z_i$.

### 3.3.2 DISCRIMINATOR $f_\theta(\mathcal{D}_i|x)$

Using the definition and the Bayesian theorem, $\log f_\theta(\mathcal{D}_i|x)$ can be written as follows:

$$\log f_\theta(\mathcal{D}_i|x) = \log \int \frac{p_\theta(\mathcal{D}_i|z)p_\theta(z|x)}{p_\theta(\mathcal{D}_i)} dz = \log \int \frac{p_\theta(z|\mathcal{D}_i)p_\theta(z|x)}{p_\theta(z)} dz$$

$$= \log \int \frac{\mathcal{N}(z|\mu_i, \sigma^2)\mathcal{N}(z|\mu_\phi(x), \sigma^2)}{\mathcal{N}(z|0, I)} dz = \frac{\mu_i^\mathrm{T} \mu_\phi(x)}{\sigma^2}. \tag{6}$$

The equation above indicates $\log f_\theta(\mathcal{D}_i|x)$ can be written simply as the inner product between $\mu_i$ and $\mu_\phi(x)$, and the objective can be written as follows:

$$\mathbb{E}_{i\sim[n]} \left[ y_i \log f_\theta(\mathcal{D}_i|x) \right] = \frac{\mathbf{y}^\mathrm{T} U \mu_\phi(x)}{n\sigma^2} = \alpha \mu_U^*(\mathbf{y})\mu_\phi(x), \tag{7}$$

where $U = (\mu_1, \ldots, \mu_n)^\mathrm{T}$, $\mu_U^* = \mathbf{y}^\mathrm{T} U/n$ and $\alpha = \sigma^{-2}$. Interestingly, it only requires one additional parameter $U$ except a hyperparameter $\alpha$. $U$ is named as a *domain embedding matrix*, representing the set of the domain prototypes. Domain embedding makes it possible to extend our method to infinite domains such as a continuous domain. In fact, $\mu_U^*(\mathbf{y}) \in \mathcal{Z}^*$ represents a prototype of mixed domains indicated by $\mathbf{y}$ in a *domain latent space* $\mathcal{Z}^*$, a dual space of $\mathcal{Z}$. Note that $\dim \mathcal{Z} = \dim \mathcal{Z}^*$.

### 3.3.3 REGULARIZER $p(\theta)$

The overall parameters of DualVAE is $\theta = (w, \phi, U)$, where $w$ is the encoder's , parameter$\phi$ is the decoders's parameter, and $U$ is the domain embedding matrix. While a typical VAE does not assume any distribution of $w, \phi$, $p(U)$ is set as an exponential distribution with an additional hyperparameter $\beta \in (0, \infty)$ to obtain sparse representation: $p(U) \propto \exp(-\beta\|U\|_1/\gamma)$ where $\|\cdot\|_1$ is a 1-norm, thus,

$$\log p(\theta) = \log p(U) = -\frac{\beta}{\gamma}\|U\|_1 - m \dim \mathcal{Z} \log 2 + \log \beta - \log \gamma. \tag{8}$$

As the terms except for the first are independent of $\theta$, we ignore them later as constants.

### 3.3.4 THE FINAL FORM

By putting together the prior, the discriminator, and the regularizer, the variational lower bound of the point-wise objective of DualVAE $J(\theta|x, \mathbf{y})$ can be written as a surprisingly simple form:

$$J(\theta|x, \mathbf{y}) \geq \mathcal{L}_\theta(x) + \alpha \langle \mu_\phi(x), \mu_U(\mathbf{y}) \rangle - \beta\|U\|_1, \tag{9}$$

where $\langle u, v \rangle = v^\mathrm{T} u$. Consequently, a DualVAE maximizes a duality paring $\langle \cdot, \cdot \rangle : \mathcal{Z} \times \mathcal{Z}^* \to \mathbb{R}$ between the sample latent space $\mathcal{Z} = \mathcal{Z}_\phi(\mathcal{X})$ and the domain latent space $\mathcal{Z}^* = \mathcal{Z}_U^*(\mathcal{Y})$ where

$\mathcal{Y} = \{0, 1\}^n$. Note that the objective requires only *two* additional hyperparameters in addition to the VAE. If $\alpha, \beta \to 0$, it is equivalent to a single VAE. Intuitively, $1/\alpha$ and $1/\beta$ control variance and bias of the domain embeddings, respectively.

The training algorithm of the DualVAE is shown in Algorithm 1.

---

**Algorithm 1** Variational domain adaptation through DualVAE

---

**Require:** observations $(x_j)_{j=1}^m$, batch size $M$, VAE/encoder optimisers: $g$, $g_e$, hyperparameters $\alpha, \beta$, and the label matrix $Y = (\mathbf{y}_j)_{j=1}^m$.
   Initialize encoder, decoder and domain embedding parameters: $\phi, w, U$
   **repeat**
      Randomly select batch $(x_j)_{j \in \mathcal{B}}$ of size $M$
      Sample $z_j \sim q_\phi(z|x_j) \ \forall j \in \mathcal{B}$
      $\phi, w \leftarrow g(\nabla_{\phi,w} \sum_{j \in \mathcal{B}}[\log p_w(x_j|z_j) - D_{\mathrm{KL}}(q_\phi(z|x_j)\|p(z))])$
      $\phi, U \leftarrow g_e(\nabla_{\phi,U} \sum_{j \in \mathcal{B}}[\alpha(Y_{:,j} - U^T z_j)^2 + \beta\|U\|_1])$
   **until** convergence of parameters $\theta = (\phi, w, U)$

---

## 4 EXPERIMENT

Based on an original numerical experiment in domain adaptation, we confirmed that the DualVAE learns multiple distributions both qualitatively and quantitatively. Similar to the case of the existing methods, domain adaptation was confirmed via an image-generation task in this study. First, we performed A *facial image recommendation task*, which is a content-based recommendation task for generating the preferences of users. Second, we performed the standard domain transfer task with 40 domains in CelebA (Liu et al., 2015) and we showed that DualVAE outperformed two state-of-the-art methods through GAN and VAE.

The objective of the first task was to generate an image that was preferred by a specific user. We set the input space $\mathcal{X}$ as the raw image, the prior $p(x)$ as faces, and the domain $\mathcal{D}_i$ as a user. We used the dataset of CelebA and SCUT-FBP5500 as the samples from the prior. The objective of the task was to generate samples from $p_\theta(x|\mathcal{D}_i)$, exhibiting the images that were preferred by a user. We used label $y_i \sim p(\mathcal{D}_i|x)$ as the existing dataset of SCUT-FBP5500 with 5,500 faces and 60 users for the content-based recommendation.

The purpose of the second task was to transfer samples from $p(x)$ into samples from $p_\theta(x|\mathcal{D}_i)$. We set the prior $p(x)$ as face images and the posterior $p_\theta(x|\mathcal{D}_i)$ as face images with certain attributes of CelebA. We used label $y_i \sim p(\mathcal{D}_i|x)$ as the attribute of CelebA.

The results revealed that the DualVAE successfully learned the model of the target distribution $p_\theta(x|\mathcal{D}_i)$ both quantitatively and qualitatively. Quantitatively, we confirmed that the discriminator learned the distribution by evaluating the negative log-likelihood loss, $-\log p_\theta(\mathcal{D}_i|x)$. We evaluated the samples using the *domain inception score* (DIS), which is the score for evaluating the transformation of images into multiple target domains. Notably, the DIS of the DualVAE was higher than several models. Qualitatively, we demonstrated that the image could be transferred to improve the evaluation by interpolating the image. We further exhibited several beautiful facial images that the users were conscious of by decoding each domain embedding $\mu_i$, which can be considered as the projection of the ideal from inside the users. In addition, 40 domain-transferred images using the dataset of CelebA by the proposed method was better than the images by other models.

### 4.1 DATASET

**CelebA**   CelebA(Liu et al., 2015) comprises approximately 200,000 images of faces of celebrities with 40 attributes.

**SCUT-FBP5500**   SCUT-FBP5500(Liang et al., 2018) comprises 5500 face images and employs a 5-point scale evaluation by 60 people in terms of beauty preference. The face images can be categorized as Asian male, Asian female, Caucasian male, and Caucasian female, with 2000, 2000, 750, 750 images, respectively.

## 4.2 RESULT

The quantitative result of the experiment can be demonstrated by evaluating the generated images by several models using a Domain Inception Score (DIS). Although the Inception Score (Salimans et al., 2016) is a score for measuring generated images, it can only measure the diversity of the images, and it is not for evaluating domain transfer of the images. Therefore, we proposed using a DIS, which is a score for evaluating the transformation of images into multiple target domains.

The DIS is a scalar value using the output of Inceptionv3(Szegedy et al., 2016) pretrained to output the domain label, and it is evaluated by the sum of two elements. The first is whether the domain transfer of the original image has been successful (transfer score), and the second is whether the features other than the transferred domain are retained (reconstruction score). A more detailed explanation of the DIS is provided in the appendix.

**Comparison of a DualVAE and a single-domain VAE**  A DualVAE can transform the image of the source domain into images of multiple target domains with one model. However, considering a simpler method, it is also possible to transfer the image of the source domain to the images of the multiple target domains by creating multiple models. We will call each of these models a Single Domain VAE (SD-VAE). Since an SD-VAE is a model that converts the image of one source domain to the image of one target domain, models corresponding to the number of target domains are required, and thus, 60 models required training. We demonstrated that the DualVAE performance was equal to or higher than that of the SD-VAE using the DIS. With respect to the output images of these two models, the one with a higher DIS value was considered to be capable of outputting ideal images. We calculated the DIS of 200 test images transferred by these two model. The DIS of the DualVAE was -0.0185, whereas that of the SD-VAE was -0.0282. Thus, the DIS of the DualVAE was 0.01 higher than that of SD-VAE.

**Comparison of DualVAE and several models**  The DualVAE was compared with several models capable of performing image-to-image translations for multiple domains using a single model. In this experiment, only the celebA dataset and the attributes of the dataset were used as the domain. Also, the input image was resized to $128 \times 128$. In each model, the dimension of the latent variable and the learning rate were randomly changed, the DIS was calculated several times, and the average and the standard deviation were obtained. The DualVAE obtained a higher DIS than the other models.

Table 1: Average DISs for three domain adaptation methods employing random hyperparameter search, which demonstrates DualVAE outperforms several models based on the DIS as the hyperparameter search is not necessary. Typical generated images are shown in the Appendix.

| Method | 5 domains | 10 domains | 20 domains | 40 domains |
|---|---|---|---|---|
| CVAE(Kingma et al., 2014) | -0.055±0.011 | -0.108±0.017 | -0.112±0.007 | -0.152±0.006 |
| UFDN (Liu et al., 2018) | 0.251±0.011 | 0.160±0.013 | 0.075±0.008 | -0.002±0.003 |
| StarGAN (Choi et al., 2017) | 0.239±0.261 | -0.094±0.346 | 0.068±0.188 | 0.050±0.032 |
| DualVAE | **0.278**±0.026 | **0.180**±0.011 | **0.163**±0.025 | **0.140**±0.020 |

## 4.3 VISUALIZATION OF DOMAIN TRANSFER

We transferred the images by interpolating between the original and the target domain images. We calculated the following vector $\mathbf{w}_i$:

$$\mathbf{w}_i = \mathbf{z} + \lambda \boldsymbol{\mu}_i. \tag{10}$$

Here, $\mathbf{w}_i$ was constrained by giving it the same norm as $\mathbf{z}$ to retain as much of the original features as possible.

By changing $\lambda$ and decoding $\mathbf{w}_i$, five images were determined to represent unideal to ideal reconstructions for each of the three sample users ($i = 14, 18,$ and $32$), and interpolation was performed to approach the ideal image $\mathbf{x}_i$ in Figure 3. In addition, we have visualized transferred images of the 40 attributes by the proposed method and other models in Figure 4.3. Although StarGAN and UFDN

retained the characteristics of the original image considerably, it was qualitatively understood that domain transfer was not good especially when the number of domains was large like 40 attributes.

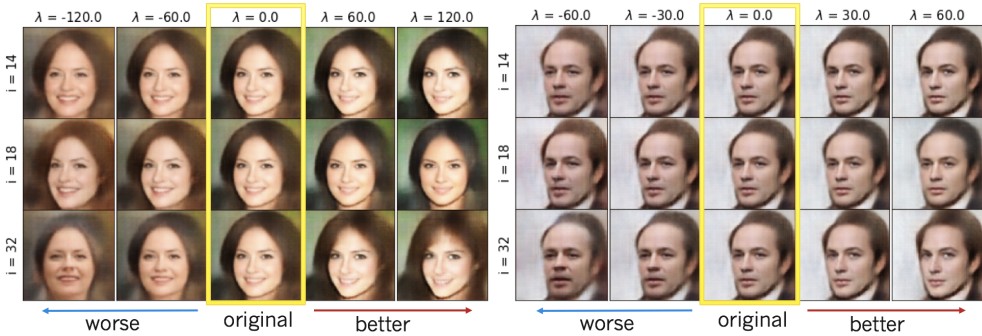

Figure 3: Images obtained from our model by decoding $\mathbf{w}_i (i = 14, 18,$ and $32)$ while changing the value of $\lambda$. The reconstructed images are present in the center.

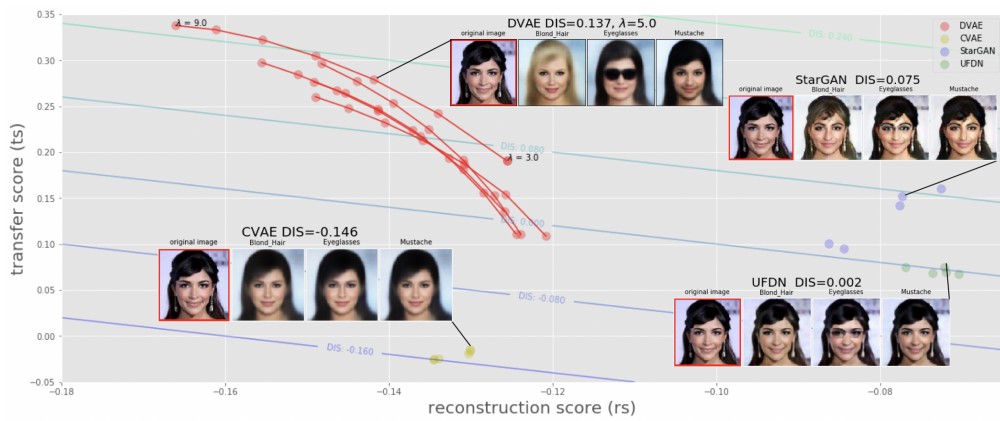

Figure 4: Scatter plot of DVAE, UFDN, StarGAN and CVAE when we change several parameters. Different color denotes different models. All 40 domains transferred images are in subsection C.1.

## 5 CONCLUSION

Variational domain adaptation, which is a unified framework for learning multiple distributions in a single network, is proposed in this study. Our framework uses one known source as a prior $p(x)$ and binary discriminator $p(\mathcal{D}_i|x)$, thereby discriminating the target domain $\mathcal{D}_i$ from the others; this is in contrast with the existing frameworks in which samples undergo domain transfer through deep generative models. Consequently, our framework regards the target as a posterior that is characterized through Bayesian inference, $p(x|\mathcal{D}_i) \propto p(\mathcal{D}_i|x)p(x)$. This was exhibited by the proposed DualVAE. The major feature of the DualVAE is domain embedding, which is a powerful tool that encodes all the domains and the samples obtained from the prior into normal distributions in the same latent space as that learned by a unified network through variational inference. In the experiment, we applied our framework and model to a multi-domain image generation task. celebA and face image data that were obtained based on evaluation by 60 users were used, and the result revealed that the DualVAE method outperformed StarGAN and UFDN.

Several directions should be considered for future research. First, we intend to expand DualVAEs for learning in complex domains, such as high-resolution images with several models, for example, glow(Kingma & Dhariwal, 2018). Second, we will perform an experiment to consider wider domains with respect to beauty. We expect that our proposed method will contribute to society in a number of ways and will help to deal with the paradigm of multiple contexts—multimodal, multi-task, and multi-agent contexts.

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

## A    LATENT SPACE

We visualized the latent space $\mathcal{Z}$ of VAE and DualVAE. VAE differs from DualVAE methodology because evaluation regression is not conducted during training. For each model, we can achieve 5500 latent vectors of 63 dimensions by encoding 5500 images from SCUT-FBP5500. We obtained a scatter plot after using UMAP (McInnes & Healy, 2018) to reduce the number of dimensions to two. The average score is indicated by colors ranging from red to blue. As can be observed from the UMAP of DualVAE, the gradient of the score is learned, and it represents the user vector(domain embedding vector) in Figure 5.

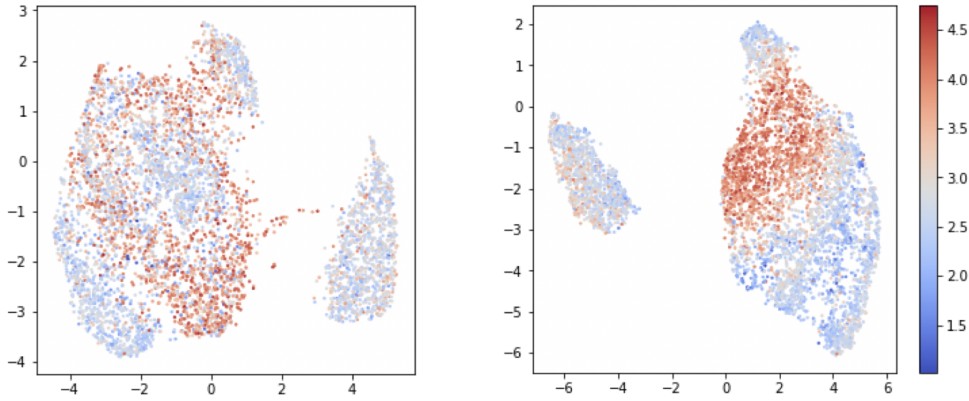

Figure 5: Latent visualization of VAE (left) and DualVAE (right) demonstrates that DualVAE learns a good prior to model the domains. The heat map indicates the mean score of all the users.

## B    DOMAIN INCEPTION SCORE (DIS)

Although the Inception Score (Salimans et al., 2016) is a score for measuring generated images, it can only measure the diversity of the images, and it is not for evaluating domain transfer of the images. Therefore, we proposed using a DIS, which is a score for evaluating the transformation of images into multiple target domains.

DIS is a scalar value, and it is evaluated by the sum of two elements. The first is whether the domain transfer of the original image has been successful (transfer score), and the second is whether the features other than the transferred domain are retained (reconstruction score).

We calculated the DIS using Algorithm 2. First, we assumed that there were N domains and we knew which domain each image belongs to. We fine-tuned Inceptionv3 (Szegedy et al., 2016) using images X as inputs and domains as outputs. To enable the model to classify the images as the domains, we replaced the last layer of the model in a new layer which had N outputs. Second, we transferred test

images into N domains using Equation 10 and loaded the transferred images into the Inceptionv3 pretrained above. Through this process we got $N \times N$ matrix for every original image, because one image was transferred into N domains and each domain image was mapped to N-dim vector. We then mapped the original image into N-dim vector using Inceptionv3, and subtracted this vector from each row of the abobe $N \times N$ matrix. We named this matrix M. The key points are (1) the diagonal elements of M should be large because we transferred the original image into the diagonal domains, and (2) the off-diagonal elements of M should be small because the transferred images should preserve original features as possible. In a later subsection, we will directly visualize these two elements and evaluate models.

---

**Algorithm 2** Domain Inception Score (DIS)

---

**Require:** observation $x \in \mathcal{X}$, Inceptionv3 $f$, domain transfer model $m$.
   $x' \leftarrow m(x)$
   $M \leftarrow f(x') - f(x)$
   ts $\leftarrow$ average(diag(M))
   rs $\leftarrow -$average(abs(notdiag(M)))
   DIS $\leftarrow$ ts+rs

---

In the Algorithm, abs denotes taking the absolute value, diag denotes taking the diagonal elements of the matrix, notdiag denotes taking the non-diagonal elements, avg denotes taking the mean of multiple values.

## C ADAPTATION OVER MANY DOMAINS

This section shows further results of Table 1, the experimental result for domain adaptation over 40 domains made from CelebA. In the experimental setting above, we use attributes in CelebA as a domain, the setting is used by several studies with domain adaptation (Choi et al., 2017). The result shows DualVAE only learns 40 domains in *one* network, which indicates DualVAE is an easy way to learn over 10 domains.

Next, we show several experimental results when we change the parameters of the models. Because StarGAN uses GAN, the learning rate parameter is not robust, thus the learning is not conducted well. Moreover, celebA has 40 domains which are too many for StarGAN, and this can also be considered as one of the reasons that learning is not conducted well. Because reconstruction is conducted well, rs in Algorithm 2 becomes larger than that of DualVAE. On the other hand, domain transfer is not conducted properly, ts in Algorithm 2 becomes extremely small compares to that of DualVAE. Therefore, as we can see from Table 1, DIS becomes a very small value.

### C.1 CELEBA

DVAE DIS=0.28, ts=-0.14, rs=0.14, $\lambda$=5.0

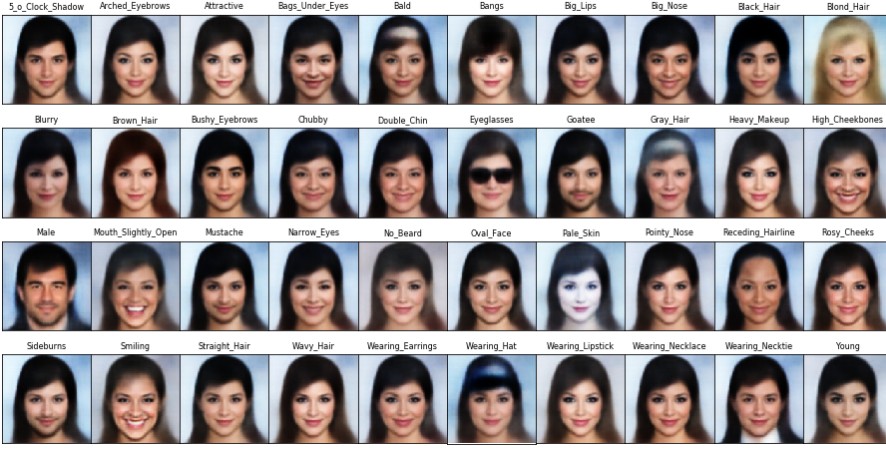

(a) DualVAE

CVAE DIS=-0.01, ts=-0.13, rs=-0.14

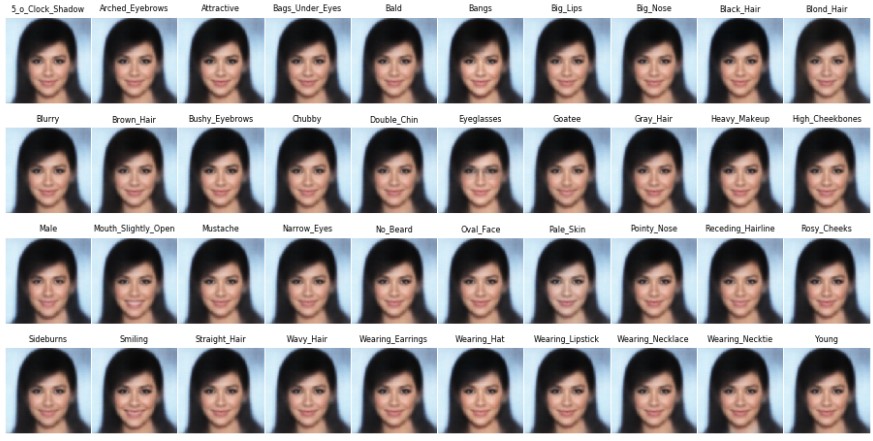

(b) CVAE

UFDN DIS=0.00, ts=0.07, rs=-0.07

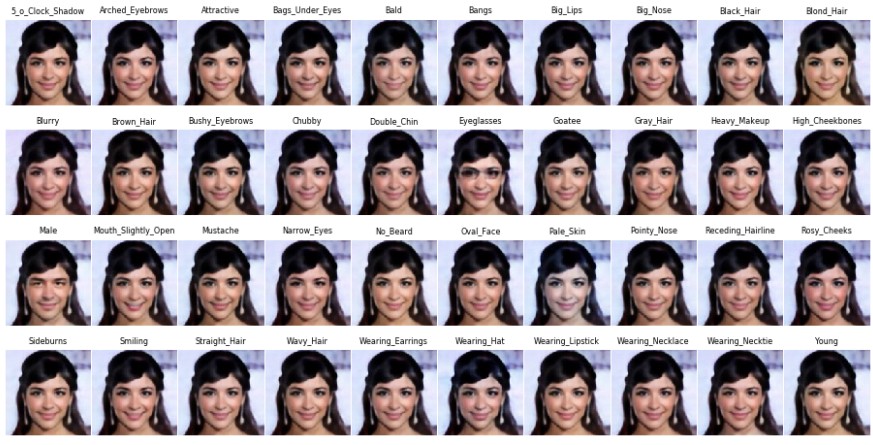

(c) UFDN

StarGAN DIS=0.07, ts=0.15, rs=-0.08

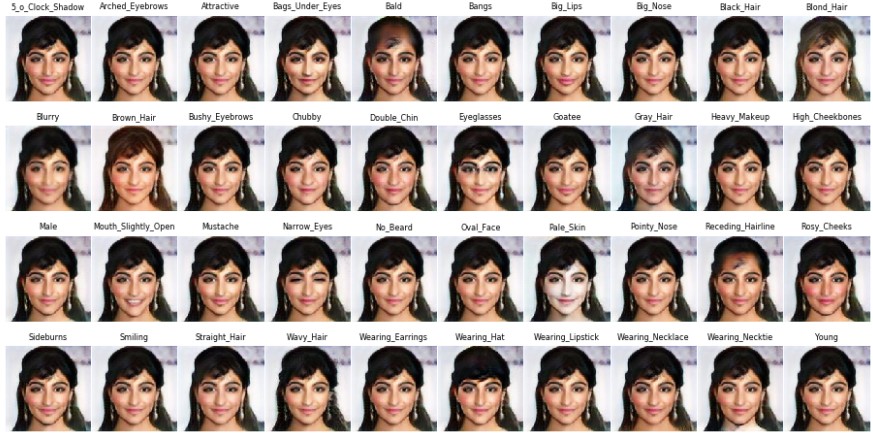

(d) StarGAN

Figure 6: Comparing domain transfer by several methods. (a) Since the image is blurry compared to StarGAN, although the original features change significantly, domain transfer is still conducted properly. (b) Although the characteristics of the original image are well preserved, domain transfer and reconstruction is not conducted. (c) Although the characteristics of the original image are well preserved, domain transfer is not conducted well. (d) Keeping the characteristic and being able to transfer a small amount of the domain.

## C.2 MNIST (10 DOMAINS)

Next, we conduct domain transfer experiments using the MNIST dataset. In this experiment, we demonstrated that it is possible to transfer the image into another label (domain), while not compromising the style of the original image. We also plotted the relation with DIS when labels are sparse. Moreover, we showed in subsection I.1 it is possible to transfer to another domain step by step.

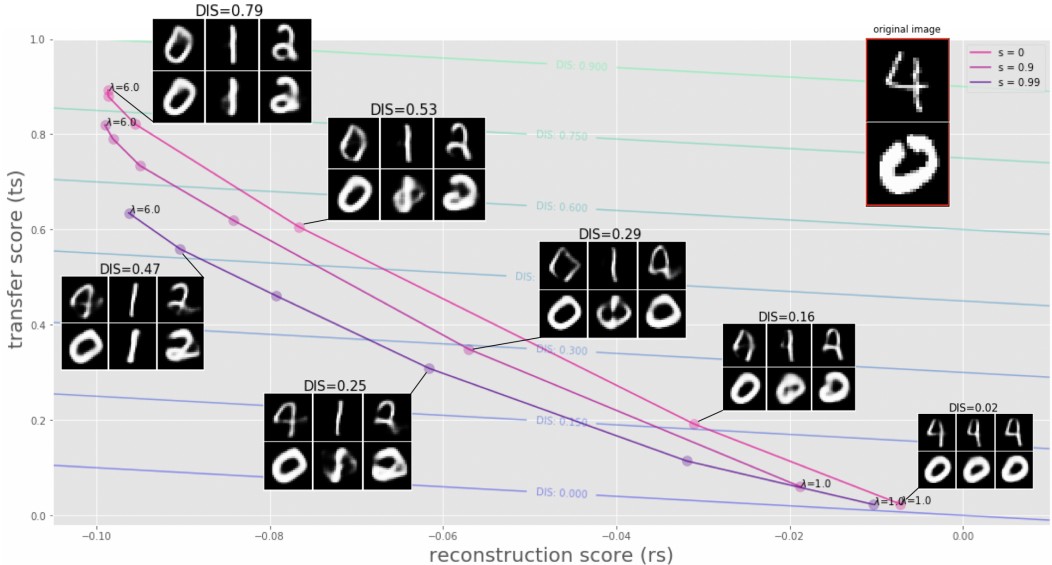

Figure 7: Scatter plot of the missing ratio of MNIST's label and DIS of the DualVAE. Variable s is the missing ratio. The original image is shown in the top right of the figure. The labels of the original images are transformed to zero, one and two. The vertical axis is ts of Algorithm 2, the horizontal axis is rs of Algorithm 2. DIS grows in the upper right corner.

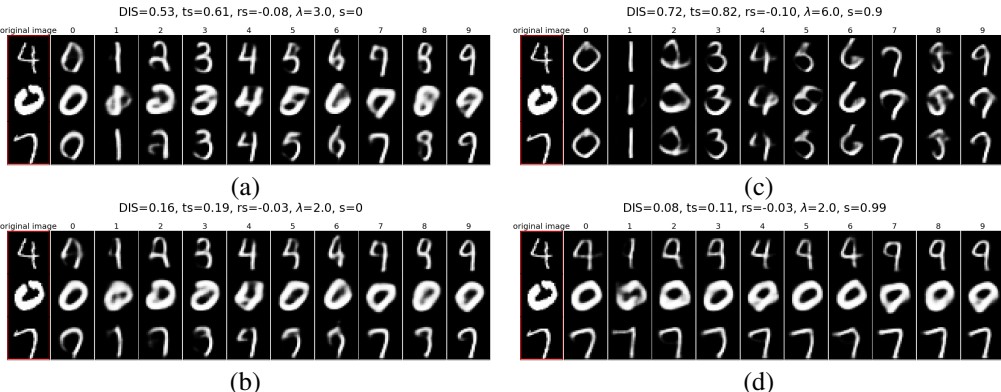

Figure 8: Domain transfer by varying $\lambda$. (a) Good example: domain transfer to different label is successful while keeping the characteristic of the reconstruction image. (b) Bad example: although the characteristic of reconstruction images are kept, domain transfer to different labels is not enough. (c) Bad example: although domain transfer to different label is successful, the characteristic of reconstruction images is lost a little bit. (d) Bad example: domain transfer to different labels is not successful.

## D  DOMAIN EMBEDDINGS

By reducing the dimensions of the 60 domain embedding vectors from 63 to 2 using UMAP(McInnes & Healy, 2018), the domain embedding vectors were visualized by means of a scatter plot. Furthermore, $x_i$ was visualized by decoding samples from the domain distribution.

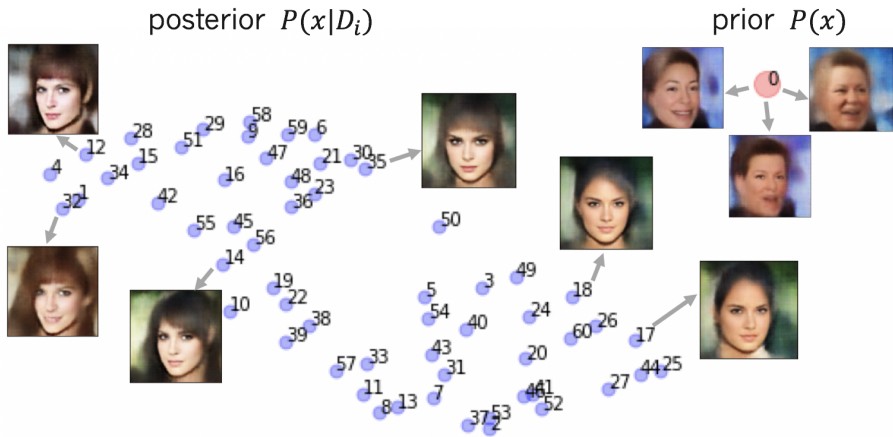

Figure 9: Scatter plot of the domain embedding vectors, and several decoded images of the samples from each domain. Six $z_i$ from the target domain distribution and output $x_i$ were decoded. Furthermore, $z_0$ from the source domain data distribution and output $x_0$ was also decoded.

## E  DOMAIN MIXING

In this chapter, we show it is possible to conduct arithmetic operations among domains. For example, suppose we learned the embedding vector of a charming image domain for each single person. We can output the charming image for the group of people as an entity without learning simply by taking the average value of the domain embedding vectors. Denote Community preference as $f_I$, personal evaluation model as $f_i (= \boldsymbol{\mu}_i^{\mathrm{T}} \mathbf{z}(x))$,

$$f_I(x) = \frac{1}{|I|} \sum_{i \in I} f_i(x) = \frac{1}{|I|} \sum_{i \in I} \boldsymbol{\mu}_i^{\mathrm{T}} \mathbf{z}(x) = \bar{\boldsymbol{\mu}}^{\mathrm{T}} \mathbf{z}(x), \tag{11}$$

where, $\bar{\boldsymbol{\mu}} = (1/|I|) \sum_{i \in I} \boldsymbol{\mu}_i$ , which is the average of domain embedding vectors. Moreover, i is the index denoting the domain (person), I is the number of domains, and z(x) is the latent vector of image x.

As shown in Equation 11, since the domain embedding vectors are linearly functional, by taking the inner product of the average of these vectors $\bar{\boldsymbol{\mu}}$ and the latent vector $\mathbf{z}$, the average of personal evaluation (evaluation of the community) can be obtained.

Therefore, by substituting $\boldsymbol{\mu}_i$ for $\bar{\boldsymbol{\mu}}$ in Equation 10, we can reconstruct the face images with high a high degree of community evaluation. We reconstructed for higher (and lower) evaluation using 10 face images from both genders. Each image enjoys higher evaluation to the right. We can see that gradually the caving becomes deep, the beard disappears, the eyes become bigger and the outline becomes sharp Figure 10.

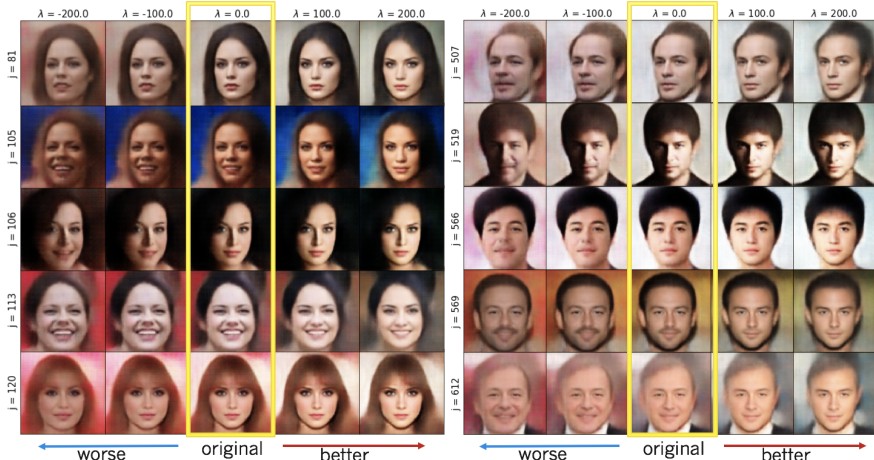

Figure 10: Images decoded to get closer (or further) to the average of domain embedding vectors. The middle is the reconstructed original image.

# F    CONNECTION TO THE HISTORY OF MATRIX FACTORIZATION

The section tells the proposed method, DualVAE, is a natural generalization from probabilistic Matrix Factorization (PMF)(Mnih & Salakhutdinov, 2008), proposed in ten years ago.

## F.1    PROBABILISTIC MATRIX FACTORIZATION (PMF)

PMF is used in several application area, mainly collaborative filtering algorithm, which are typical recommendation algorithms. PMF learns the user matrix $U \in \mathcal{R}^{K \times N}$ and the item matrix $V \in \mathcal{R}^{K \times J}$ that can restore the evaluation matrix. Here, $r_{ij}$ is the evaluation value of item j by user i, the evaluation matrix is denoted as $R \in \mathcal{R}^{I \times J}$. Moreover, the column vector of the user matrix $U$ and the item matrix $V$ are denoted as $\mathbf{u}_i, \mathbf{v}_j$ respectively. K is the dimension of these vectors, N is the number of users, J is the number of items. $I_{ij}$ is the indicator function that takes the value 1 when evaluation $r_{ij}$ exists and 0 otherwise.

The log likelihood of PMF is

$$\log p(R|U, V, \sigma^2) = \sum_i \sum_j I_{ij} \log \mathcal{N}(r_{ij}|\mathbf{u}_i^T \mathbf{v}_j, \sigma^2). \tag{12}$$

Our objective is to find the $\mathbf{u}_i$, $\mathbf{v}_j$ that maximizes the above.

**Relationship to DualVAE**    DualVAE is an end-to-end coupling of VAE and PMF. We could see DualVAE as PMF extended to a generative model. $\mathbf{u}_i$ in Equation 12 corresponds to the domain embedding vector in DVAE, $\mathbf{v}_j$ corresponds to the latent vector in DVAE, $r_{ij}$ corresponds to the likelihood that item j belongs to domain i.

## F.2    EXPERIMENTAL ANALYSIS

### F.2.1    EFFECT OF END-TO-END COUPLING

We experimentally show that the DualVAE outperformed the non-end-to-end coupling. We compared two models. One is the model trained to regress evaluation of the image end-to-end by calculating inner product of hidden representation of VAE and domain embedding (DVAE). The other is the model which learns hidden representation of VAE followed by learning to regress evaluation by inner product like above (VAE-PMF). We used SCUTFBP-5500 Figure 17 dataset, and validated it into 5000 images with 60 evaluators and 500 test images with 60 evaluators. We quantitatively compared these two models in terms of Root Mean Square Error (RMSE) of model prediction and reconstruction error of test images. The result suggests that DualVAE achieved a much smaller

RMSE. Moreover, though DualVAE constrained its hidden representation to regress evaluation, the reconstruction error was almost the same as VAE-PMF. This suggests that DualVAE can generate as clear images as vanilla VAE.

Table 2: compare DualVAE with VAE

| Method | RMSE | Reconstruction loss |
|---|---|---|
| VAE + PMF | 0.423 | $8.19 \times 10^4$ |
| DualVAE | **0.356** | $8.20 \times 10^4$ |

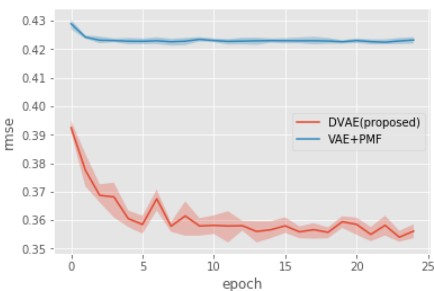
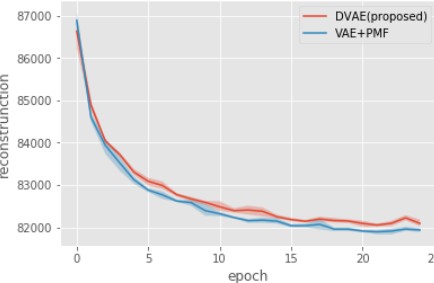

Figure 11: RMSE and Reconstruction loss. DualVAE is far superior to VAE in classification accuracy, and there is almost no difference in reconstruction error between them.

### F.2.2 ROBUSTNESS TO SPARSITY

In addition to generalization capability, another benefit from PMF is robustness to sparsity as PMF is robust to a matrix with many missing values. We will experimentally demonstrate that DualVAE is also robust with respect to sparse labels. We calculate the rs and ts when applying Algorithm 2 on 160 celebA test images, and plot the below figure when we change the missing ratio of celeA's domain labels and the $\lambda$ in Equation 10.

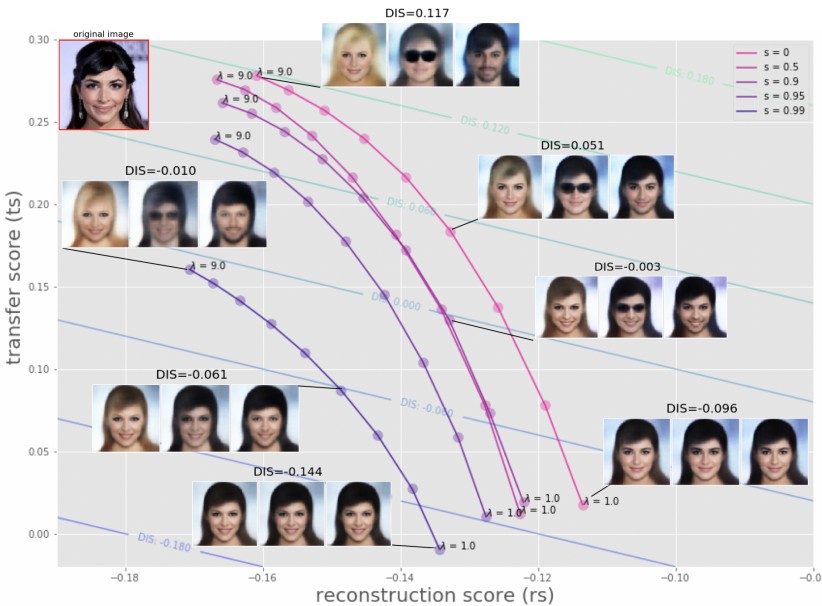

Figure 12: Scatter plot of the missing ratio of celeA's label and DIS of DualVAE. Variable s is the missing ratio. The original image is shown on the left top of the figure. The attributes of the original images are transformed to blond hair, eyeglasses and mustache. The vertical axis is ts of Algorithm 2, the horizontal axis is rs of Algorithm 2. DIS grows in the upper right corner.

DIS=0.05, ts=0.18, rs=-0.13, λ=4.0, s=0

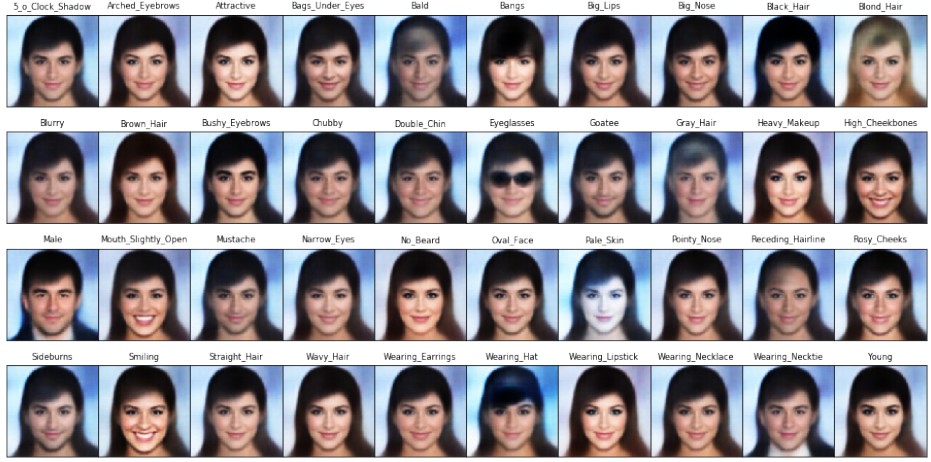

(a) $s = 0$. Keeping the characteristic and being able to domain transfer.

DIS=0.03, ts=0.17, rs=-0.14, λ=4.0, s=0.9

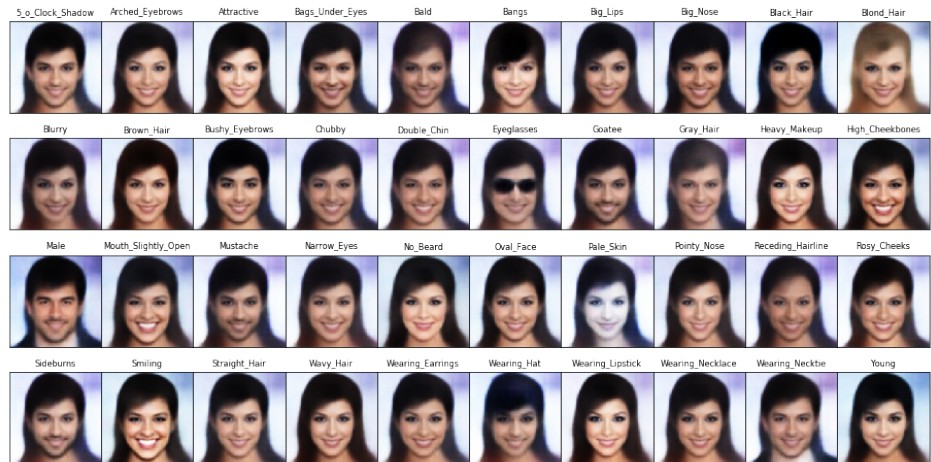

(b) $s = 0.9$. Although the sparseness of the labels is high, domain transfer is was still conducted rather well.

DIS=-0.01, ts=0.16, rs=-0.17, λ=9.0, s=0.99

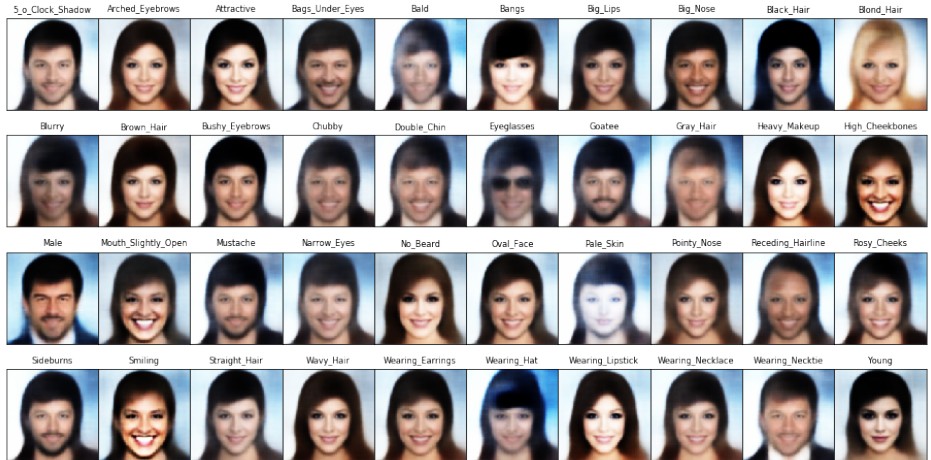

(c) $s = 0.99$ (bad example). Image quality is poor, domain transfer is not conducted properly.

Figure 13: Sparsity analysis through various sparsity.

From Figure 13, keeping the characteristic of the upper right plots, it is possible to conduct domain transfer at the same time. Moreover, the method is strong on the sparseness of domain labels, and DIS does not drop even when 90 of the labels are missing.

On the other hand, we show that StarGAN is not as robust as DualVAE with respect to sparseness. When 90 of domain labels are missing, StarGAN cannot learn at all and generates identical images.

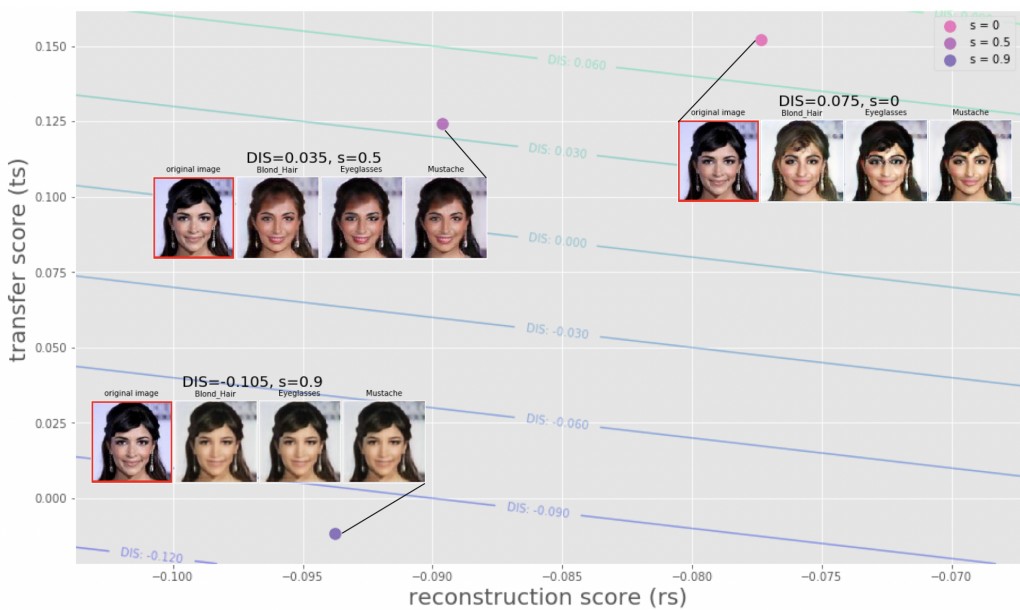

Figure 14: Scatter plot of the missing ratio of celeA's label and DIS of StarGAN. Variable s is the missing ratio. The vertical axis is ts of Algorithm 2, the horizontal axis is rs of Algorithm 2. DIS grows at the upper right corner.

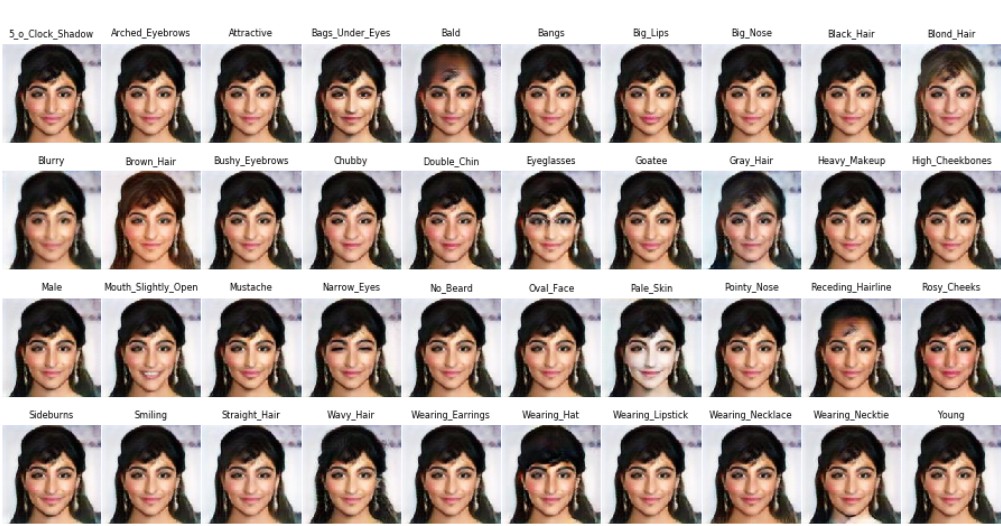

(a) $s = 0$. Keeping the characteristic and being able to transfer a small amount of the domain.

DIS=-0.11, ts=-0.01, rs=-0.09, s=0.9

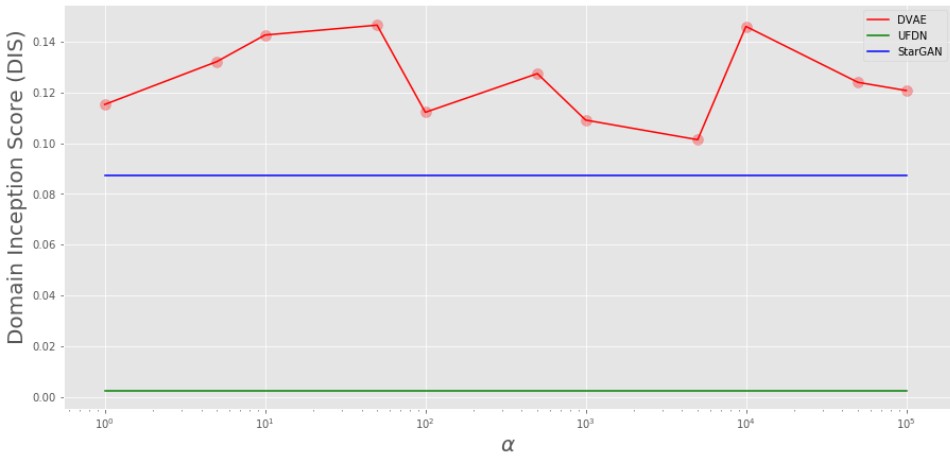

(b) $s = 0.9$. All identical images are generated, and domain transfer is not properly conducted.

Figure 15: Sparsity analysis through StarGAN.

## G   FURTHER EXPERIMENTS WITH VARIOUS HYPERPARAMETER ($\alpha$)

We conducted a comparison experiment with the existing methods when changing $\alpha(= \sigma^{-2})$ in Equation 9. Here, the number of domains was set to 40. As you can see from the results below, it turns out that the performance of DualVAE is robust to $\alpha$.

Figure 16: Plot of DVAE, UFDN, and StarGAN when we change alpha. The performance of DVAE is rubust to $\alpha$ and DVAE outperforms the existing methods based on the DIS.

## H   MODEL DETAILS

The section shows three models used in tasks of domain adaptation over three types of domains: environment, attribute and class.

**Environment**   First, we describe the experimental setting for domain transfer to the ideal image of each individual. We assumed that the beauty criterion required for evaluating the facial images de-

pends on the gender of a person in the target image. Therefore, we added the gender information to the images. For this purpose, we applied CGAN (Mirza & Osindero, 2014) to VAE. We normalized the scoring in $[-1, 1]$ to accelerate the learning. Subsequently, we considered the specific model structure of DualVAE. Both the input and output images were RGB images, $x \in R^{256 \times 256 \times 3}$. We used convolution networks for the encoder and stride 2 for convolution and no pooling. Convolution, batch normalization(Ioffe & Szegedy, 2015), and LeakyReLU were repeated four times and were subsequently connected to fully connected layers. Further, after batch normalization and LeakyReLU layers, a 63-dimensional latent variable was obtained. The decoder exhibited a completely symmetric shape with deconvolution layers instead of convolution layers. Furthermore, as the gender attribute, we set 0 as female and 1 as male. We added an image $x \in R^{256 \times 256 \times 1}$ comprising 0 or 1 data as the input of the encoder and a scalar of 0 or 1 for gender to the latent variable, which was the input to the decoder. The detailed structure is in Structure A of Table 3. We optimized DualVAE on SCUT-FBP5500. Because there were no face evaluation data in celebA, we only used it to optimize VAE. Learning was alternatively realized using these two datasets. We show the image example of SCUT-FBP5500 (Liang et al., 2018). From Figure 17, we can see the evaluation value depends on each person.

**Attribute**  Next, in comparative experiment with several models, domain transfer was performed with only celebA data and domain number of 40, 20, 10, and 5. We experimented with several parameters of the models. In particular, the dimensions of the latent variable and the learning rates were randomly selected. Both the input and output images were RGB images, $x \in R^{128 \times 128 \times 3}$. The detailed structure is in Structure B of Table 3.

**Class**  Finally, we describe the experimental setting of domain transfer in the MNIST dataset. This experimental result is stated in the subsection C.2. Both the input and output images were gray images, $x \in R^{28 \times 28 \times 1}$. The detailed structure is in Structure C of Table 3.

Table 3: The model structures of DualVAE used in our experiment. Conv stands for Convolution, Deconv stands for Deconvolution, FC stands for Full Connected, and numbers in each parenthesis are input and output channels. The kernel sizes of convolution are all $4 \times 4$, and stride sizes of that are all two. Also, Batch2d represents batchnormalization in two dimensions, Batch1d represents batchnormalization in one dimension, and LReLU stands for LeakyReLU.

| Preference | Attribute | Class |
|---|---|---|
| Conv(3+1, 32) | Conv(3, 64) | Conv(1, 64) |
| Batch2d,LReLU | Batch2d,LReLU | Batch2d,LReLU |
| Conv(32, 64) | Conv(64, 128) | Conv(64, 128, 4, 4) |
| Batch2d,LReLU | Batch2d,LReLU | Batch2d,LReLU |
| Conv(64, 128) | Conv(128, 256) | FC(128*7*7, 1024) |
| Batch2d,LReLU | Batch2d,LReLU | Batch1d,LReLU |
| Conv(128, 256) | FC(256*16*16, 1024) | FC(1000, 100*2) |
| Batch2d,LReLU | Batch1d,LReLU | FC(100, 1024)+FC(100, 10) |
| FC(256*16*16, 1024) | FC(1024, dim*2) | Batch1d,LReLU |
| Batch1d,LReLU | FC(dim, 1024)+FC(dim, 40) | FC(1024, 128*7*7) |
| FC(1024, 63*2) | Batch1d,LReLU | LReLU |
| FC(63+1, 1024)+FC(63+1, 60) | FC(1024, 256*16*16) | Deconv(128, 64) |
| Batch1d,LReLU | LReLU | Batch2d,LReLU |
| FC(1024, 256*16*16) | Deconv(256, 128) | Deconv(64, 1) |
| LReLU | Batch2d,LReLU | Sigmoid |
| Deconv(256, 128) | Deconv(128, 64) | |
| Batch2d,LReLU | Batch2d,LReLU | |
| Deconv(128, 64) | Deconv(64, 3) | |
| Batch2d,LReLU | Sigmoid | |
| Deconv(64, 32) | | |
| Batch2d,LReLU | | |
| Deconv(32, 3) | | |
| Sigmoid | | |

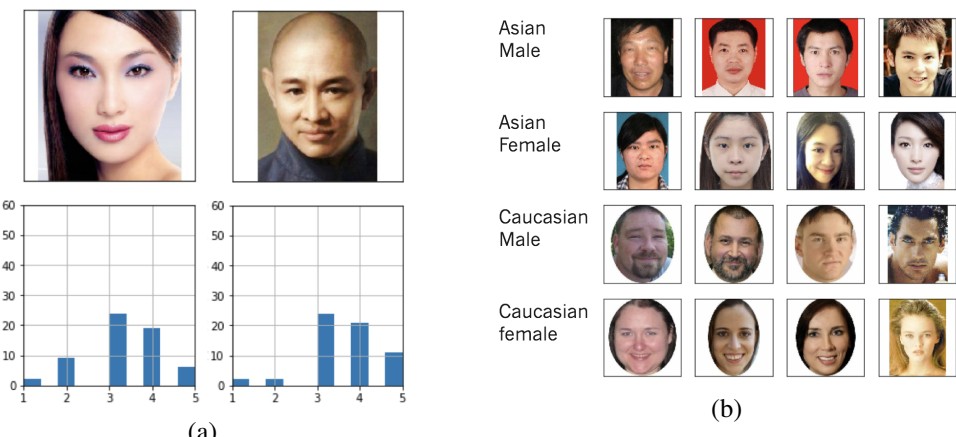

Figure 17: (a) Distribution of the respective scores within [1, 5] rated by 60 people. (b) The images of SCUT-FBP5500.

# I FURTHER EXAMPLES OF DOMAIN TRANSFER THROUGH DUALVAE

The results below shows result from domain adaptation performed by DualVAE by randomly-sampled images from two datasets: MNIST and CelebA.

## I.1 MNIST (10 DOMAINS)

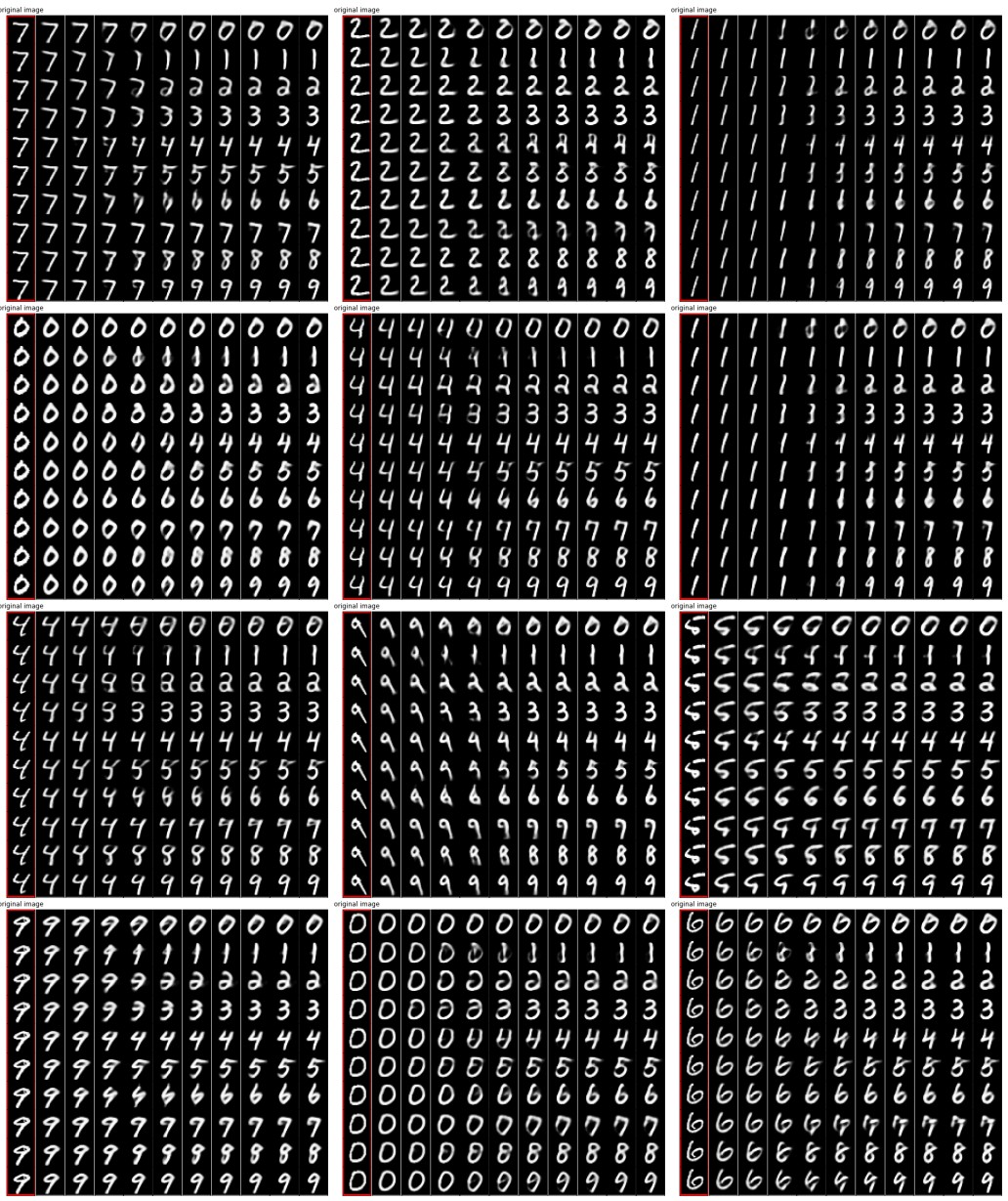

Figure 18: DualVAE stably transfers samples across 10 domains while domain-irrelevant features (e.g., style) are kept.

## I.2 CELEBA (40 DOMAINS)

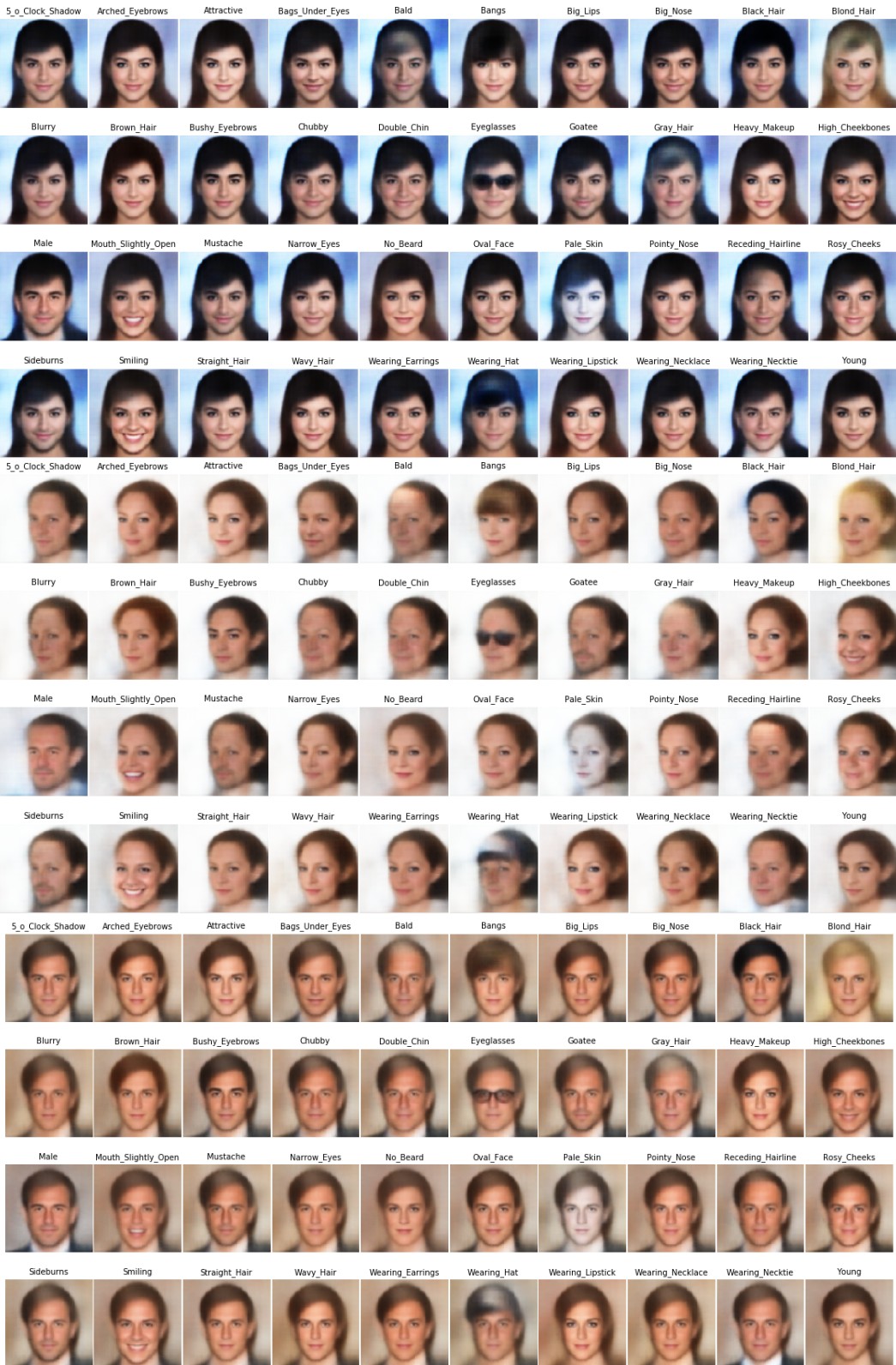

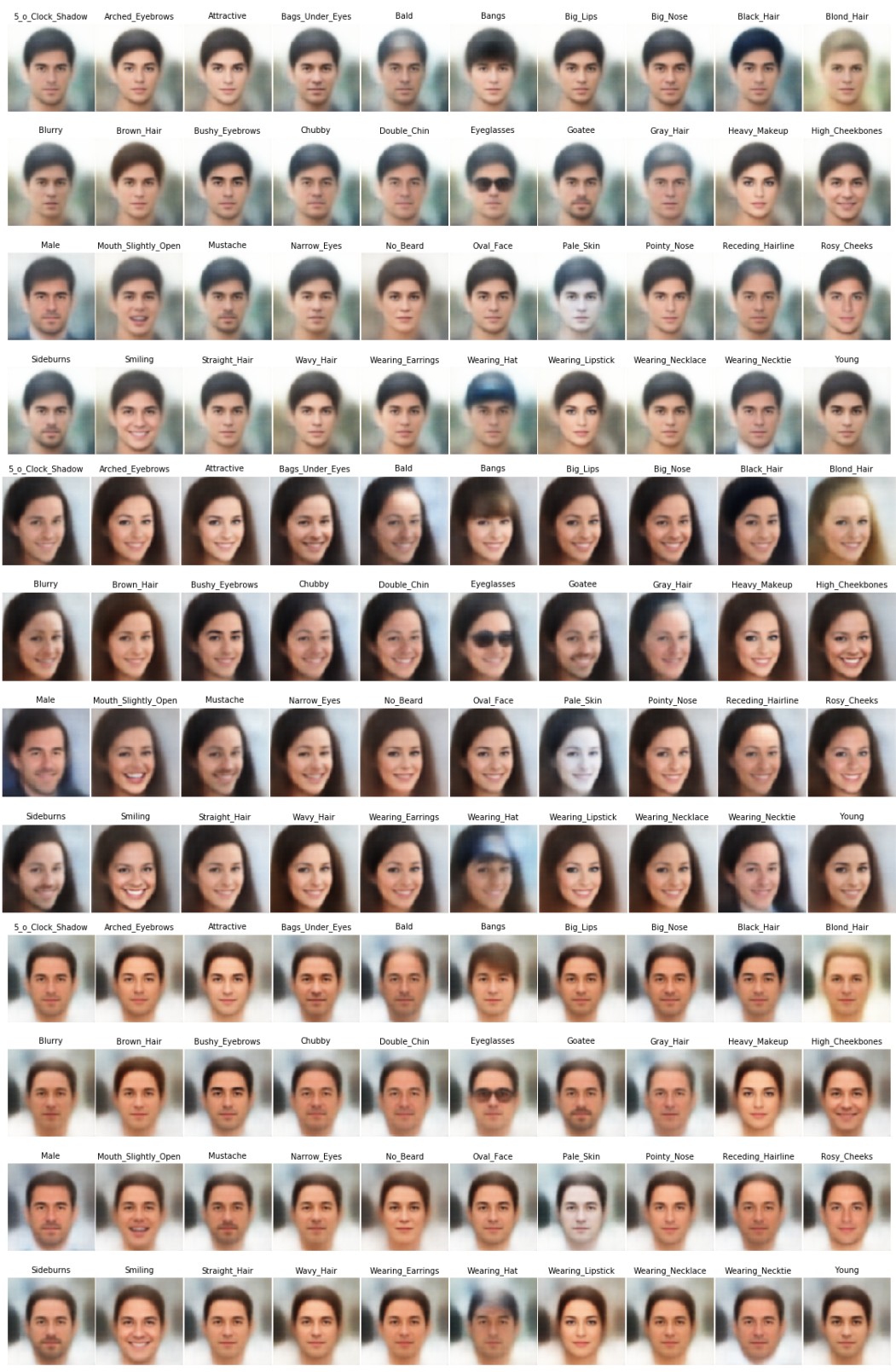

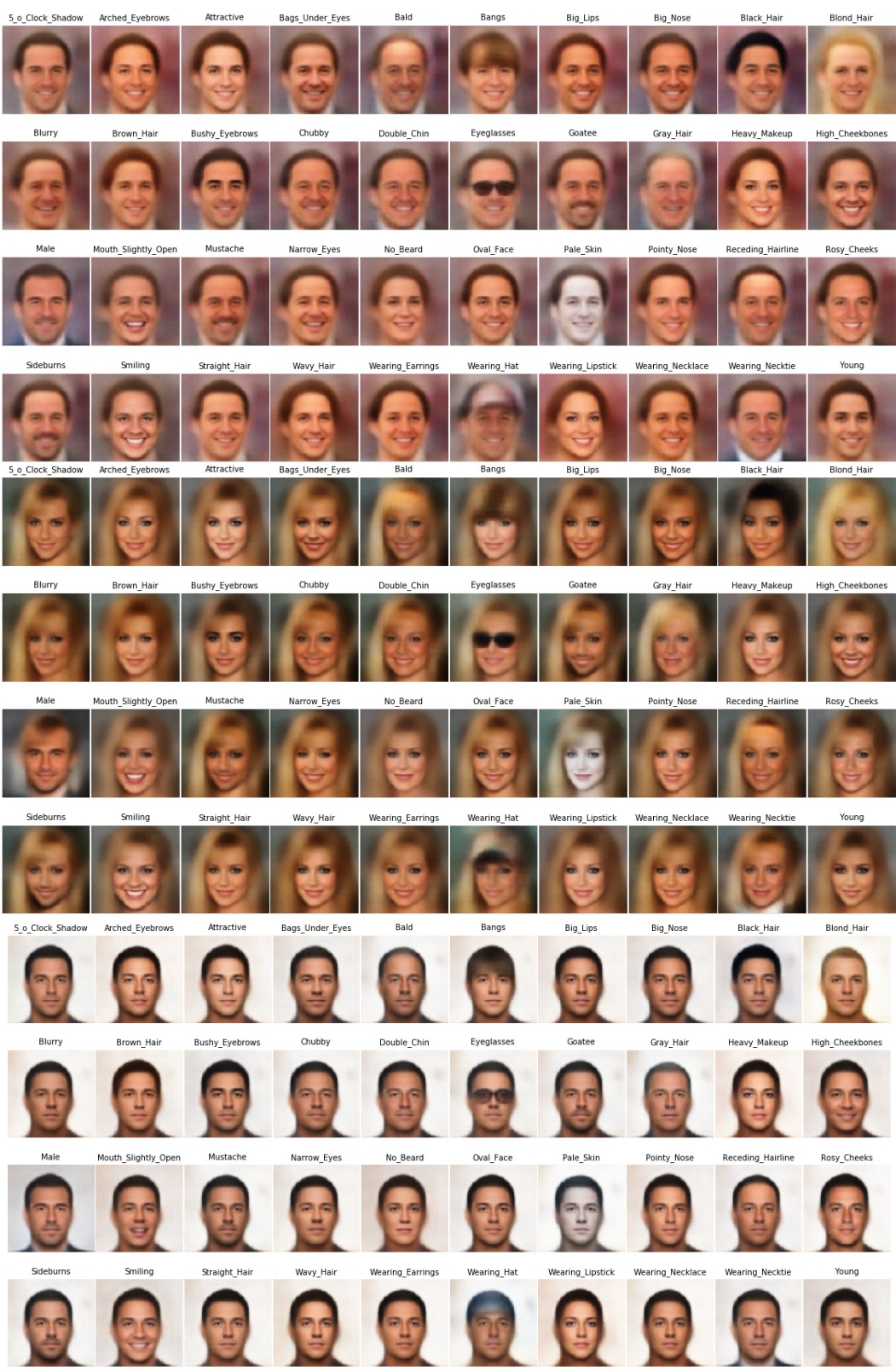

