# OpenReview forum: "Variational Domain Adaptation"
_ICLR.cc/2019/Conference_

### Official Review · AnonReviewer2 · 2018-11-01

**Rating:** 5
**Confidence:** 3

**Review:**

In this paper, the authors propose a variational domain adaptation framework for learning multiple distributions through variational inference. The proposed framework assumes a prior, and models each domain as a posterior. A multi-domain variational auto-encoder is then proposed to implement the concept of multi-domain semi-supervision. Experimental studies are done to show the effectiveness of the proposed framework.

This paper does not deal with the conventional domain adaptation problem as many existing domain adaptation works do. It focuses on the adaptation task of data generation. Here are some comments:
(1)	It would be better to clarify the adaptation task by giving a concrete real-word example in the introduction. Specifically, you may want to specify what the source and target tasks are, and what the assumption you have made on the source and target tasks is.
(2)	In the abstract and introduction, you state that a source domain is regarded as a prior, and target domain is regarded as posterior. From the Method section, I am not sure whether this is a valid statement. In my understanding, equation (1) is the KL summation of all the domains. The following derivation assumes that the data of all the domains draw a distribution p(x) (which is the prior), and the data of each domain has a specific distribution p^(i)(x) (which is the posterior).  Do you assume that all the domains from D_i to D_n are target domains? Then, what are the source domains?
(3)	From eq.(2) to eq.(3), why p(D_i) = \lamda_i is assumed? Is p(D_i) related to the number of the instance in D_i?
(4)	In the prior part of eq.(3), it should have a p(D_i|x) before log p_\theta(x), right? Where is f(\hat_{D}|x), in the first line of page 4, used? What are the optimizers: g and g_e?
(5)	Regarding the experimental studies, what do you want to conclude from the visualization of the domain embeddings? It would be better to give more discussion, analyses or observation for the visualization. For the comparison result with StarGAN, could you elaborate the experimental settings for each method? Could you give more explanation on why MD-VAE outperforms StarGAN. Furthermore, are there any other state-of-the-art baselines that can be compared?

Overall, I think this is an interesting paper. However, there are some unclear parts need to be further clarified. The experimental studies are a litter weak in the sense that (1) it needs more discussion and analyses on the results; and (2) more baselines need to be compared.

---

> ### Author Response · Authors · 2018-11-16
> **Response to reviewer2**
>
> Thanks for your feedback.
>
> > (2) In the abstract and introduction, you state that a source domain is regarded as a prior, and the target domain is regarded as a posterior. From the Method section, I am not sure whether this is a valid statement. In my understanding, equation (1) is the KL summation of all the domains. The following derivation assumes that the data of all the domains draw a distribution p(x) (which is the prior), and the data of each domain has a specific distribution p^(i)(x) (which is the posterior). Do you assume that all the domains from D_i to D_n are target domains? Then, what are the source domains?
>
> In fact, the image set of the target domain p(x|D_i) was contained in the source domain p(x).
> Specifically, p(x) is a whole face image set, and p(x|D_i) is a face image set that people (i) like.
> To consider domain transfer when the image set of the source domain and the image set of the target domain are independent, we prepared two target domain sets p(x|D_1) and p(x|D_2).
> You can see that p(x|D_1) should be a source domain set and p(x|D_2) a target domain set.
>
> > (3) From eq.(2) to eq.(3), why p(D_i) = \lamda_i is assumed? Is p(D_i) related to the number of the instance in D_i?
>
> Yes, \lambda_i is the percentage of D_i.
>
> > (4) In the prior part of eq.(3), it should have a p(D_i|x) before log p_\theta(x), right?
>
> Yes. Thank you for the observation.
>
> > Where is f(\hat_{D}|x), in the first line of page 4, used?
>
> We did not use it.
>
> > What are the optimizers: g and g_e?
>
> The optimizer g is for both the VAE encoder and decoder; g_e is the optimizer for the VAE encoder. Both the optimizers are Adam.
>
> > Regarding the experimental studies, what do you want to conclude from the visualization of the domain embeddings? It would be better to give more discussion, analyses or observation for the visualization.
>
> Visualizing the domain embeddings, we showed that the original image set p(x) can be transformed into the image set p(x|D_i) of multiple domains.
> However, we think that there were some unclear parts; therefore, we changed the image to a clearer image with a graph of quantitative comparison with other models. Please see the image on p. 8.
>
> >  For the comparison result with StarGAN, could you elaborate the experimental settings for each method? Could you give more explanation on why MD-VAE outperforms StarGAN.
>
> In the comparison experiment with the existing method, the test images of the CelebA domain transferred by the methods were compared using DIS and changing the parameter five times.
> Since the CelebA dataset had 40 kinds of attributes, we changed the number of attributes, such as 5, 10, 20, 40, and performed domain transformation. Please see the results on p. 7.
>
> > Furthermore, are there any other state-of-the-art baselines that can be compared?
>
> We added experiments of UFDN (NIPS, 2018) to the experimental results (p. 7) of the body.
> The reason for choosing UFDN is that SOTA of the domain transfer is StarGAN in the method based on GAN, but it is UFDN in the method based on VAE.

---

### Official Review · AnonReviewer3 · 2018-11-02
**A preference learning generative model (in deep setting), with somewhat unintuitive setting and weak experimental evaluation**

**Rating:** 4
**Confidence:** 5

**Review:**

1) Summary of the paper:

The paper brings up a relatively new problem of learning a generative model for multiple domains. The domains, D1,...,Dn, may refer to person-specific preferred images, for instance, and they focus on how to build generative models P(x|Di), which represents a set of images preferred by subject i.

They assumed a specific setup where one can access domain classifiers P(Di|x), but not the samples from P(x|Di). It is a bit odd: actually they worked mostly on a special (relatively new) dataset named "SCUT-FBP-5500", which seems to contain labeled samples, (x,D1,...,Dn) -- then, obviously we can access x|Di as well as Di|x. Of course, this type of fully labeled dataset is small-sized.

Their approach is basically to partition the latent space by the domains D1,...,Dn. They utilize the standard VAE model which is shared across the domains, and introduce domain-specific latent priors P(z|D_i) which are Gaussians. The learning is essentially a combination of the VAE learning and the latent prior learning, where the latter is done by enforcing the generated samples x from each Di to be consistent with the domain classifier P(Di|x). This strategy sounds reasonable enough.

One issue lies in the latent prior learning (ie, optimization of (3)). Since they need to evaluate P(Di|x), x is limited to the labeled samples, namely those from the (small-sized) SCUT-FBP-5500 dataset only. So although they wrote expectation wrt p(x) in (3), the p(x) cannot be a large dataset like the CelebA dataset as they intended, but p(x) is limited to a small dataset like SCUT-FBP. The large samples from p(x) are only exploited in the VAE learning part.

The experimental evaluation is weak: evaluated on only one dataset, compared with just standard VAE and StarGAN which are not aimed for the particular problem setup the authors are considering.

At least, they may be able to compare it with a baseline approach, e.g., using the samples from p(x|D_i) available from the SCUT dataset (small though), one can learn encoder/decoder models for each D_i.

2) Strengths:

Relatively unique problem (but unusual and unintuitive setup) and a reasonable approach.

3) Weak points:

-The writing is sloppy. It doesn't read very well, and difficult to follow. Contains many typos.

-Weak in experimental evaluation and comparison with other (baseline) approaches.

-There appears to exist identity change in many of face image preference examples.  This is unexpected.  I would be more inclined to believe that personal preferences are about appearance (style) features rather than identify.  Yet most examples in Fig.6 indicate the opposite.

- Writing would benefit from laying out intuition beyond both the model and the experimental results.

---

> ### Author Response · Authors · 2018-11-16
> **Response to reviewer3**
>
> Thanks for your comments.
>
> > They assumed a specific setup where one can access domain classifiers P(Di|x), but not the samples from P(x|Di). It is a bit odd: actually they worked mostly on a special (relatively new) dataset named "SCUT-FBP-5500," which seemed to contain the labeled samples (x,D1,...,Dn). Then, obviously we could access x|Di as well as Di|x. Of course, this type of fully labeled dataset is small in size.
>
> Since the samples from p(x|D_i) were not large to clearly generate images, we needed to obtain a sample from p(x) instead of p(x|D_i).
>
> > One issue lies in the latent prior learning (ie, optimization of (3)). Since they need to evaluate P(Di|x), x is limited to the labeled samples, namely those from the (small-sized) SCUT-FBP-5500 dataset only. So although they wrote expectation wrt p(x) in (3), the p(x) cannot be a large dataset like the CelebA dataset as they intended, but p(x) is limited to a small dataset like SCUT-FBP. The large samples from p(x) are only exploited in the VAE learning part.
>
> Please explain this again because we are unable to understand the meaning of “the p(x) cannot be a large dataset like the CelebA dataset.”
> We used the CelebA dataset as a prior distribution of facial images because clear images could not be generated using only the SCUT dataset.
>
> > The experimental evaluation was weak; it was evaluated on only one dataset, as compared with the standard VAE and StarGAN, which were not aimed for the particular problem setup that the authors were considering.
>
> In response to your suggestions, we performed the experiments again with two additional datasets:
>     1.  CelebA (40 domains)
>     2.  MNIST (10 domains)
> The experiments showed good result and the details of the results have been added in the appendix.
>
> As written in the revised paper, in the experiments using the SCUT-FBP-5500 dataset, we regarded the preference of one person as one domain.
> In additional experiments using CelebA and MNIST, we performed domain transfer of facial image attributes and numeric labels, respectively.
> We also conducted additional comparison experiments using CelebA with UFDN (NIPS, 2018), CVAE.
> UFDN was chosen because the SOTA of the domain transfer was StarGAN in the method based on GAN but it was UFDN in the method based on VAE.
> Moreover, since the proposed method is a model that extends CVAE, we also compared it with the original CVAE.
> Please see the updated results on page 7.
>
> > At least, they may be able to compare it with a baseline approach, e.g., using the samples from p(x|D_i) available from the SCUT dataset (small though), one can learn encoder/decoder models for each D_i.
>
> Yes, we experimentally confirmed that the DualVAE is more accurate than the single domain VAEs (SD-VAEs) learned independently in each domain by an experiment using SCUT-FBP-5500 (60 domains).
> This is stated in the original paper in the SDVAE vs MDVAE paragraph (p.7).
>
> > There appears to be identity changes in many of the face image preference examples. This is unexpected. I would be more inclined to believe that personal preferences are about appearance (style) features rather than identify; yet, most examples in Fig. 6 indicate the opposite.
>
> Since the image in Fig. 6 was not explained, the explanation was added in section E (p.14).
> This figure shows that by averaging the embedding of domains, DualVAE generates images that are preferred for multiple people.
> However, the more preferred the image, the more the identity will change as you have pointed out.
> Therefore, it is important to continuously adjust the parameters to the extent that the identity does not change.

---

### Official Review · AnonReviewer1 · 2018-11-03
**Unclear contribution**

**Rating:** 4
**Confidence:** 3

**Review:**

- To this reviewer’s understanding, the proposed method is very similar or equal to the conditional VAE. The only difference comes from the way of involving the condition information during training.  This should be clarified and further, it is necessary to compare with the conditional VAE in the experiments, rather than the vanilla VAE.

- The proposed method uses a predefined and fixed value of the variance $\sigma^{2}$, which is very informative and should be estimated from data in inference. Basically, there is no specification on this value in their experiments.

- In a similar perspective, how the results changes according to the variation of the value $\sigma$.

- It is not intuitive how significant the improvement of 5% in PIS. It would be good to provide the intuitive understanding of the improvement.

---

> ### Author Response · Authors · 2018-11-16
> **Response to reviewer1**
>
> Thanks for your feedback.
>
> > The proposed method is similar to or equal to the conditional VAE. The only difference is the way the condition information is involved during training.
>
> While the key idea of the proposed method has been used by CVAE, there are no studies that have argued for the relation of the CVAE to domain adaptation.
> Therefore, our main contribution is bridging CVAE and domain adaptation using DualVAE.
>
> > From a similar perspective, we can see that the results change according to the variation of the value $\sigma$.
>
> Below are the results of one of the additional experiments, which indicates that the performance is robust to $\sigma$.
>
> Method                 |      DIS
> -------------------------------------------
> StarGAN                 |      0.087
> UFDN                     |      0.002
> DualVAE (α=1)       |      0.115
> DualVAE (α=10)     |      0.143
> DualVAE (α=10^2) |      0.112
> DualVAE (α=10^3) |      0.109
> DualVAE (α=10^4) |      0.146
>
> α=\sigma^{- 2}; the number of domains: 40
> Since variance of the prior p(x) is 1, we set α >= 1.
> Additional details are listed in Fig. 16, Appendix G (p. 19)
>
> > It is not intuitive how significant the improvement of 5% in PIS. It would be good to provide the intuitive understanding of the improvement.
>
> PIS = reconstruction score + domain transfer score.
> Please see Figure 4 (p. 8) and Appendix B to intuitively understand PIS.
> (In the paper, we changed the name of PIS to DIS).
> We compared DualVAE with StarGAN, UFDN, and CVAE, and showed the relation between DIS and the result of domain transfer using each method.

---

### Public Comment · ~Christian_B_Goldberg1 · 2018-11-05
**Very clever idea. Is the source code publicly available?**

The paper proposes a quite clever trick with inner product and variational autoencoder to address the important research issue of convergence in domain adaptation. It shows an experimental result for transfer across quite large (-60) domains.

I'd like to use the technique in my work because several solutions such as StarGAN did not converge although I've tried.
If you have any source code, I'm happy if you can share us it publicly/privately.

---

> ### Author Response · Authors · 2018-11-16
> **Response to Christian**
>
> Thank you for your comments. We intend to make the code public.

---

### Author Response · Authors · 2018-11-16
**We have rewritten the body to reflect the reviewers' suggestions**

Thank you for your review comments. We have rewritten the body to reflect your suggestions.
We have added a few more experiments in the appendix and have increased the number of pages from 12 to 25 (including the appendix).
Please note that we have renamed several terms in the body.
1.  Multi-domain VAE (MD-VAE) --> DualVAE
2.  Preferential Inception Score (PIS) --> Domain Inception Score (DIS)

---

### Public Comment · ~Yifei_Wang1 · 2019-09-17
**Your inner product trick (Eq. 6) seems to be wrong**

Check: let u_i=u_phi=0, sigma^2=2, density ratio = int{1/sqrt(8pi) dz}, which is not integerable. For general cases, even assumed integrability, it can also be shown the equality does not hold.

To get your result, you should assume p(z) to be N(0, sigma^2) rather than N(0, I). If insisting your assumptions, you would end up with a quadrature rather than only linear product involved.

Furthermore, you assume p(z)=N(0,I). This is not consistent with p(z|Di)=N(ui, sigma2), which makes p(z) a mixture of Gaussian distribution if D is categorical.

---

### Meta-Review · Area_Chair1 · 2018-12-12
**Additional discussion and experiments appreciated, more improvements needed**

**Confidence:** 4
**Recommendation:** Reject

**Metareview:**

This paper proposes using conditional VAEs for multi-domain transfer and presents results on CelebA and SCUT. As mentioned by reviewers, the presentation and clarity of the work could be improved. It is quite difficult to determine the new/proposed aspects of the work from a first read through. Though we recognize and appreciate that the authors updated their manuscript to improve its clarity, another edit pass with particular focus on clarifying prior work on conditional VAEs and their proposed new application to domain transfer would be beneficial.

In addition, as DIS is the main metric for comparison to prior work and for evaluation of the final approach, the conclusions about the effectiveness of this method would be easier to see if a more detailed description of the metric and analysis of the results were provided.

Given the limited technical novelty and discussion amongst reviewers of the desire for more experimental evidence, this work is not quite ready for publication.